JOURNAL OF
Neuroscience Research

# Functional brain network topology across the menstrual cycle is estradiol dependent and correlates with individual well-being

Marianna Liparoti[1]  |  Emahnuel Troisi Lopez[1]  |  Laura Sarno[2]  |  Rosaria Rucco[1,3]  |
Roberta Minino[1]  |  Matteo Pesoli[1]  |  Giuseppe Perruolo[4,5]  |
Pietro Formisano[4,5]  |  Fabio Lucidi[6]  |  Giuseppe Sorrentino[1,3,7]  |
Pierpaolo Sorrentino[3,8]

[1]Department of Motor Sciences and Wellness, University of Naples "Parthenope", Naples, Italy

[2]Department of Neurosciences, Reproductive Science and Dentistry, University of Naples "Federico II", Naples, Italy

[3]Institute of Applied Sciences and Intelligent Systems, CNR, Pozzuoli, Italy

[4]Department of Translational Medicine, University of Naples "Federico II", Naples, Italy

[5]URT "Genomic of Diabetes" of Institute of Experimental Endocrinology and Oncology, National Council of Research, CNR, Naples, Italy

[6]Department of Developmental and Social Psychology, University of Rome "Sapienza", Rome, Italy

[7]Hermitage Capodimonte Clinic, Naples, Italy

[8]Institut de Neurosciences des Systèmes, Faculty of Medicine, Aix-Marseille Université, Marseille, France

Correspondence
Giuseppe Sorrentino, Department of Motor Sciences and Wellness, University of Naples "Parthenope", 80133 Naples, Italy.
Email: giuseppe.sorrentino@uniparthenope.it

Funding information
University of Naples Parthenope "Ricerca locale" (GS)

## Abstract

The menstrual cycle (MC) is a sex hormone-related phenomenon that repeats itself cyclically during the woman's reproductive life. In this explorative study, we hypothesized that coordinated variations of multiple sex hormones may affect the large-scale organization of the brain functional network and that, in turn, such changes might have psychological correlates, even in the absence of overt clinical signs of anxiety and/or depression. To test our hypothesis, we investigated longitudinally, across the MC, the relationship between the sex hormones and both brain network and psychological changes. We enrolled 24 naturally cycling women and, at the early-follicular, peri-ovulatory, and mid-luteal phases of the MC, we performed: (a) sex hormone dosage, (b) magnetoencephalography recording to study the brain network topology, and (c) psychological questionnaires to quantify anxiety, depression, self-esteem, and well-being. We showed that during the peri-ovulatory phase, in the alpha band, the leaf fraction and the tree hierarchy of the brain network were reduced, while the betweenness centrality (BC) of the right posterior cingulate gyrus (rPCG) was increased. Furthermore, the increase in BC was predicted by estradiol levels. Moreover, during the luteal phase, the variation of estradiol correlated positively with the variations of both the topological change and environmental mastery dimension of the well-being test, which, in turn, was related to the increase in the BC of rPCG. Our results highlight the effects of sex hormones on the large-scale brain network organization as well as on their possible relationship with the psychological state across the MC. Moreover, the fact that physiological changes in the brain topology occur throughout the MC has widespread implications for neuroimaging studies.

### KEYWORDS
behavior, emotional stimuli, magnetoencephalography, posterior cingulate gyrus, premenstrual dysphoric disorder, sex hormones

Edited by Inger Sundstrom Poromaa and Junie Warrington. Reviewed by Rachel Zsido and Esmeralda Hildago-Lopez.

Marianna Liparoti and Emahnuel Troisi Lopez contributed equally to this study.

# 1 | INTRODUCTION

The brain, over the course of a lifetime, undergoes continuous and dynamic changes on multiple time scales (Sporns, 2018). Hormonal modulation induces changes in both structure and function. For example, during puberty, sex hormones contribute to morphological variations of the cortical and subcortical regions (Giedd et al., 1996; Sowell et al., 2007; van Duijvenvoorde et al., 2019) involved in sensorimotor processing, such as the thalamus and the caudate, as well as areas involved in emotion and memory processes, such as the amygdala and the hippocampus. Menopause is another example of the effects that sex hormones play on brain functions affecting both cognition (Maki & Henderson, 2016) and mood (Santoro et al., 2015).

Unlike puberty or menopause, which are processes that occur in adolescence and adulthood, respectively, the menstrual cycle (MC) is a hormone-related phenomenon that accompanies the women from puberty to menopause and repeats itself cyclically with periodical and coordinated variations of multiple hormones, such as estradiol, progesterone, follicular-stimulating hormone (FSH), and luteinizing hormone (LH). Such variations can induce a number of physical (acne, breast pain, cramps, headaches), neuro-vegetative (sleep and eating disorders) (Guida et al., 2020; Yen et al., 2018), and psychopathological changes (anxiety, depression, moodiness) (Parry & Haynes, 2000).

A large number of women suffer from sex hormone-dependent psychopathological disorders, including postpartum depression, peri-menopausal depression, and premenstrual dysphoric disorder (PMDD) (Parker & Brotchie, 2010; Payne et al., 2009). PMDD is characterized by cyclic, debilitating cognitive, somatic, and affective symptoms (depression, irritability, mood lability, anxiety) which occur during the luteal phase, abate at menses, and greatly affect quality of life (Wittchen et al., 2002). PMDD, which is now categorized as a new depressive disorder in the Diagnostic and Statistical Manual of Mental Disorders (DSM–5) (American Psychiatric Association), affects approximately 5%–8% of women of reproductive age. An additional 30%–40% of women suffer from the milder, yet clinically significant, premenstrual syndrome (PMS), that also impact the quality of life (Ryu, 2019). However, a multitude of women report cycle-related emotional symptoms. Tschudin et al. reported that 57% of the women of childbearing age experience a mild degree of "anger/irritability" or "tearfulness/mood swings" in the premenstrual phase (Tschudin et al., 2010). According to several studies, the prevalence of at least one premenstrual symptom during the woman's reproductive life may be much higher, reaching up to 90% (Dennerstein et al., 2012).

The association of low self-esteem and anxiety, depression, and psychiatric symptoms in general is well known (Paxton et al., 2006). Self-esteem enables subjects to cope more effectively with stressful events (Taylor et al., 2008). Several studies suggest that low self-esteem may be regarded as a subclinical manifestation of depressive mood (Dedovic et al., 2014; Leary et al., 1995; Orth et al., 2008). The association between MC and self-esteem has been proposed. It was found that self-esteem was lower in the premenstrual

## Significance

The increased betweenness centrality in the right posterior cingulate gyrus during the peri-ovulatory phase of the menstrual cycle, which correlates with changes in both the estradiol blood levels and the psychological status, demonstrates for the first time that the brain networks topology is affected by the estradiol levels and that, in turn, is correlated with the affective state. These findings may have implication in understanding the pathophysiological basis of sex hormones related diseases such as the menstrual dysphoric disorder or the premenstrual syndrome.

period in women with premenstrual syndrome (Bloch et al., 1997; Taylor, 1999). Hill and Durante found, in a population of students, that self-esteem was higher during the late luteal phase (Hill & Durante, 2009). However, in a population with similar characteristics, Edmonds et al. failed to find an association between self-esteem and MC (Edmonds et al., 1995).

Research on the emotional condition across the MC has been based on the notion of well-being, defined as the absence of depressive and/or anxious symptoms. However, evidence suggests that a positive psychological condition entails more than the mere absence of negative mood (Clark & Watson, 1991). Accordingly, the idea that the psychological well-being must be framed in a multidimensional construct has gained acceptance (Kahneman, 1999). In particular, recent studies argue that well-being is best described taking into account elements encompassing both the "hedonic" and the "eudaimonic" dimension, the former being associated with pleasure/displeasure perceptions and changing along with life experiences (Headey, 2006), the latter being associated with homeostatic mechanisms and relatively stable. For an exhaustive review see (Ryan & Deci, 2001). Ryff (1989) claimed that the multidimensionality of the concept of psychological well-being can be conceptualized in six core dimensions (i.e., autonomy, environmental mastery, personal growth, positive relation with others, purpose in life, and self-acceptance) and developed scales to measure accordingly (Ryff, 2014).

Due to the above-mentioned physiological cyclical fluctuations, both with respect to hormone levels and emotional state, the MC can be exploited to study the relationship between sex hormone level, emotional changes and functional brain correlates. Using functional magnetic resonance (fMRI), Petersen et al. (2014) demonstrated the influence of sex hormones during follicular and luteal phases on two different functional networks, namely the anterior portion of the default mode network (aDMN) and the executive control network (ECN). In detail, comparing the brain networks in the two phases, they found that during the follicular phase, the connectivity of both the left angular gyrus within the aDMN and the right anterior cortex within the ECN was increased. Arélin et al. (2015) investigated the associations between ovarian hormones and eigenvector centrality (EC) in fMRI-based functional networks across the

MC, and found a positive correlation between progesterone and EC in the dorsolateral prefrontal cortex, the sensorimotor cortex, and the hippocampus, suggesting progesterone as a regulatory of areas involved in memory. More recently, through a multimodal analysis, changes in connectivity of subcortical networks across the MC have been demonstrated (Hidalgo-Lopez et al., 2020). The authors found in the luteal phase decreased intrinsic connectivity of the right angular gyrus in the DMN, heightened EC for the hippocampus, and increased amplitude of low-frequency fluctuations for the caudate. However, both Hjelmervik et al. (2014) and De Bond et al. (2015) did not find changes in rs-fMRI brain connectivity in different phases of the MC. The influence of sex hormones on brain connectivity was also studied through resting state electroencephalography (EEG) measurements, which provides lower spatial but higher temporal resolution as compared to fMRI. Brötzner et al. (2014) have associated the alpha frequency oscillations with the MC phases and the hormone levels, and found that the power in the alpha frequency peaked during the luteal phase and the lowest alpha power during the follicular phase, with the change negatively correlated with the estradiol levels, suggesting that the latter modulates the resting state activity in the alpha band.

In the last decades, new techniques and higher available computational power made it possible to analyze brain activity non-invasively at the whole-brain level typically applying graph theory, where nodes of the graph represent brain areas, and edges represent the inter-regional functional correlations (Bullmore & Sporns, 2009; Sporns, 2018). This approach can be applied to neurophysiological signals, such as the magnetic fields associated with electrical neural activity. Magnetoencephalography (MEG) is a non-invasive neurophysiological brain imaging method measuring the magnetic fields induced by the electrical activity in the brain, having both an excellent temporal (~1 ms) and, when coupled with spatial models (Wilson et al., 2016), good spatial (2–5 mm) resolution.

In this explorative study we hypothesized that the brain network rearranges periodically along the MC as a function of the levels of sex hormones and that these changes may be associated with modifications of the psychological status, even in the absence of overt clinical signs of anxiety and/or depression. To test our hypothesis, we used MEG to investigate the brain topology in early follicular, peri-ovulatory, and mid-luteal phases, in 24 healthy, naturally cycling women without pre-menstrual symptoms and with no signs of anxiety and/or depression. Specifically, we estimated the links between areas by assuming synchronization as a mechanism of communication (Buzsáki et al., 2013), and applied a novel metric, the phase linearity measurement (PLM) (Baselice et al., 2019), to estimate the degree of synchronization. Then, the minimum spanning tree (MST) was computed from each (frequency specific) adjacency matrix (Stam, 2014; Tewarie et al., 2015) and a set of its topological properties were estimated. Additionally, blood samples were collected to determine the hormone levels of estradiol, progesterone, LH, and FSH. Furthermore, we used a multilinear model to predict the topological parameters from the hormone blood levels. Finally, the brain topological modifications were correlated with the corresponding changes in hormone levels, as well as to the psychological status. The correlation between the hormonal levels and the psychological condition along the MC was investigated as well.

## 2 | METHODS

### 2.1 | Participants

Twenty-six strictly right-handed, native Italian speaker, heterosexual women with a regular MC, were recruited. We included women who did not make use of hormonal contraceptives (or other hormone regulating medicaments) during the last 6 months before the recording and who had not been pregnant in the last year. Furthermore, they did not use habitually drugs or medicine which could affect the central nervous system and they did not consume alcohol, tobacco, and/or coffee, 48 hr prior to the MEG recordings. Finally, we included women without history of neuropsychiatric diseases and premenstrual dysphoric/depressive symptoms, the latter checked by an expert gynecologist. To check for mood and/or anxiety symptoms, the Beck Depression Inventory (BDI) (Beck et al., 1996) and Beck Anxiety Inventory (BAI) (Beck & Steer, 1990) were used with a cut-off below 10 and 21, respectively. To control for influence of circadian rhythm, the time of testing varied no more than 2 hr among testing sessions. Along the experimental sessions, two females were excluded from the study because the BDI test value became higher than the cut-off, therefore the data analysis was carried out on a sample of 24 females. The subjects' characteristics are shown in Table 1.

### 2.2 | Experimental protocol

At enrolment, all participants signed a written consent form. All the procedures strictly adhered to the guidelines outlined in the Declaration of Helsinki, IV edition. The study protocol was approved by the local ethic committee (University of Naples Federico II; protocol n. 223/20). The women were tested in three different time points of the MC, that is, in the early follicular phase (cycle day 1–4, low estradiol and progesterone, T1), during the peri-ovulatory phase (cycle day 13–15, high estradiol and low progesterone, T2) and in the mid-luteal phase (cycle day 21–23, high estradiol and progesterone, T3). We annotated the self-reported last menstrual period, the MC length

**TABLE 1** Subject characteristics

| Demographic and anthropometric features (N = 24) | |
|---|---|
| Age (years) | 26.6 ± 5.1 |
| BMI (kg/m$^2$) | 21.6 ± 2.1 |
| Education (years) | 17.3 ± 2.7 |
| Menstrual cycle duration (days) | 28.4 ± 1.5 |
| Pregnancy | 1/24 |

*Note:* Data are given as mean ± standard deviation.

Abbreviation: BMI, body mass index.

**TABLE 2** Sex hormone assay

| | Sex hormone blood levels (N = 24) | | | | | |
| --- | --- | --- | --- | --- | --- | --- |
| | Early follicular (T1) | Peri-ovulatory (T2) | Mid luteal (T3) | $p_{FDR}$ value (T1 vs. T2)[†] | $p_{FDR}$ value (T2 vs. T3)[†] | $p_{FDR}$ value (T1 vs. T3)[†] |
| LH (mIU/ml) | 5.4 ± 2.3 | 16.1 ± 11.8 | 6.0 ± 4.0 | <0.001 | <0.001 | NS |
| FSH (mIU/ml) | 7.3 ± 1.4 | 7.7 ± 3.0 | 3.9 ± 1.2 | NS | <0.001 | <0.001 |
| Progesterone (ng/ml) | 0.3 ± 0.1 | 1.1 ± 0.8 | 5.7 ± 2.6 | <0.001 | <0.001 | <0.001 |
| Estradiol (pg/ml) | 33.9 ± 12.1 | 134.3 ± 70.6 | 97.4 ± 39.7 | <0.001 | <0.05 | <0.001 |

*Notes:* Analysis of variance (ANOVA, †) for each hormone (luteinizing hormone (LH, mIU/ml), follicular stimulant hormone (FSH, mIU/ml), progesterone (ng/ml) and estradiol (pg/ml)) estimated in 24 women (N = 24) at the early follicular (T1), peri-ovulatory (T2) and mid-luteal (T3) phases of the MC. The post hoc analysis between MC time points (T1 vs. T2, T2 vs. T3, and T1 vs. T3) was performed using a paired sample t test. Data are given as mean concentrations ± standard deviation. Significance p value: < 0.05, < 0.01, < 0.001.

Abbreviations: FDR, false discovery rate; NS, no significant.

and the date of the predicted onset of the next menses. To achieve greater accuracy in the estimation of the ovulation, the backward-counting method was applied. This is an indirect counting method that estimates ovulation by subtracting 14 days from the next predicted period onset. The date of the next period was then confirmed in all included participants (Dixon et al., 1980; Gildersleeve et al., 2013). Moreover, all the participants included in the study had normal hormonal blood levels (according to the local reference values, reported below) at the three time points, including estradiol and LH in the peri-ovulatory phase and progesterone in the mid-luteal phase. At each of the three time points along the cycle all subjects underwent the following examinations: MEG recording, blood sampling for the hormone assay, and psychological evaluation. During the early follicular phase, a transvaginal pelvic ultrasonography examination was performed. After the last MEG recording, a structural magnetic resonance imaging (MRI) was performed. Two subjects refused to execute the MRI scan and consequently the template was used for sources reconstruction. To control for a possible session effect, women were randomized according to the cycle phase at the first session. The subjects (N = 6) that did not have hormonal values in the reference range for each phase, were recorded (and tested) again in the subsequent cycles.

## 2.3 | Ultrasound examination

All participants underwent a transvaginal pelvic ultrasonography during the early follicular phase. Scans were performed using a 4–10 MHz endocavitary transducer (GE Healthcare, Milwaukee, WI). Patients were in lithotomy position with empty bladder. The uterus and both ovaries were visualized. The uterus was scanned using longitudinal and transverse plane, endometrial thickness was measured at the widest point in the longitudinal plane. Follicle number and diameters were assessed for each ovary. The presence of abnormal findings, such as endometrial polyps, myomas, ovarian cysts, or other adnexal masses was addressed. None of the enrolled patients presented abnormal findings and endometrial thickness and follicle diameters were consistent with the menstrual phase.

## 2.4 | Hormone assays

Each participant underwent venous blood sampling during the three hormonal phases of the MC. All women were asked to respect a 12-hr fast before blood collection. Whole blood samples were collected in S-Monovette tubes (Sarstedt), containing gel with clotting activator in order to facilitate the separation of the serum from the cellular fraction, according to predetermined standard operating procedure (Tuck et al., 2010). To this aim, samples were centrifuged at 4,000 rpm for 10 min, then the serum was collected, aliquoted in 1.5 ml tubes (Sarstedt), and stored at –80℃ until the analysis. Determination of estradiol (range: 19.5–144.2 pg/ml (early follicular phase), 63.9–356.7 pg/ml (peri-ovulatory phase), 55.8–214.2 pg/ml (mid-luteal phase), detection limit: 11.8 pg/ml, inter-assay coefficients of variation averaged: 1.9%, intra-assay coefficients of variation averaged: 4.9%); progesterone (range: ND–1.4 ng/ml (early follicular phase), ND–2.5 ng/ml (peri-ovulatory phase), 2.5–28.03 ng/ml (mid-luteal phase), detection limit: 0.2 ng/ml, inter-assay coefficients of variation averaged: 5.5%, intra-assay coefficients of variation averaged: 3.56%); LH (range: 1.9–12.5 mIU/ml [early follicular phase], 8.7–76.3 mIU/ml [peri-ovulatory phase], 0.5–16.9 mIU/ml [mid-luteal phase], detection limit: 0.07 mIU/ml, inter-assay coefficients of variation averaged: 2.3%; intra-assay coefficients of variation averaged: 2.5%) and FSH levels (range: 2.5–10.2 mIU/ml [early follicular phase], 3.4–33.4 mIU/ml [peri-ovulatory phase], 1.5–9.1 mIU/ml [mid-luteal phase], detection limit: 0.3 mIU/ml, inter-assay coefficients of variation averaged: 1.2%; intra-assay coefficients of variation averaged: 1.9%) were measured by Advia Centaur XT Immunoassay System analyzer (Siemens) which uses competitive (estradiol) or direct (progesterone, FSH, LH) immunoassay and for quantification of reaction uses Chemiluminescent Acridinium Ester technology. The 2.5th and 97.5th percentiles were used to form reference limits with 90% confidence intervals, as provided by assay manufacturers (McEnroe et al., 2014). The hormone blood levels at the three time points of the MC are reported in Table 2.

## 2.5 | MEG recording

The MEG system, developed by the National Research Council (CNR), Pozzuoli, Naples, at Institute of Applied Sciences and Intelligent Systems "E. Caianiello," is placed inside a shielded room (AtB Biomag UG-Ulm–Germany). The MEG is equipped with 154 magnetometers and 9 reference sensors located on a helmet (Rucco et al., 2019). Before each MEG session, four position coils were placed on the participant's head and their position, as well as that of four anatomical landmarks, was digitized using Fastrak (Polhemus®) (Lardone et al., 2018). The coils were activated and localized at the beginning of each segment of registration. The magnetic fields were recorded for 7 min, divided into two time intervals of 3′30″. The length of the recording was a trade-off between the need to have enough cleaned temporal series, and to avoid drowsiness (Fraschini et al., 2016; Gross et al., 2013). During the recordings, while the participants were sitting comfortably in an armchair in the cabin with their eyes closed, instructed not to think of something in particular, the electrocardiogram and electro-oculogram signals were also recorded (Gross et al., 2013). The data were sampled at $f_s = 1,024$ Hz and a fourth-order Butterworth IIR pass-band filter between 0.5 and 48 Hz was applied. After each session, all the subjects were checked for drowsiness during the recording period with the Karolinska sleep questionnaire (Åkerstedt et al., 1994).

## 2.6 | Data processing and source reconstruction

After the recording phase, the brain magnetic signals were cleaned through an automated process as described in our previous article (Sorriso et al., 2019). The FieldTrip software tool (Oostenveld et al., 2011), based on Mathworks® MATLAB, was used to implement principal component analysis (De Cheveigné & Simon, 2007; Sadasivan & Narayana Dutt, 1996), to reduce the environmental noise, and independent component analysis (Barbati et al., 2004), to remove physiological artifacts such as cardiac noise or eyes blinking (if present). For each participant, source reconstruction was performed for all segments through a beamforming procedure using the Fieldtrip toolbox similarly to Jacini et al. (2018). In short, based on the native MRI, the volume conduction model proposed by Nolte (2003) was applied and the Linearity Constrained Minimum Variance beamformer (Van Veen et al., 1997) was implemented to reconstruct time series related to the centroids of 116 regions of interest (ROIs), derived from the automated anatomical labeling (AAL) atlas (Gong et al., 2009; Hillebrand et al., 2016). We only considered the first 90 ROIs, excluding those corresponding to cerebellum, given that the reconstructed signal might be less reliable.

## 2.7 | Construction of brain network

After the signal had been filtered in each canonical frequency band (i.e., delta, theta, alpha, beta, and gamma—see later), the PLM (Sorrentino et al., 2019) was computed, to provide an estimate of synchronization between any two regions that is purely based on the phases of the signals, and unaffected by volume conduction. The PLM is defined as (Baselice et al., 2019):

$$PLM = \frac{\int_{-B}^{B} \left| \int_0^T e^{i\Delta\varnothing(t)} e^{-i2\pi ft} dt \right| 2 \, df}{\int_{-\infty}^{\infty} \left| \int_0^T e^{i\Delta\varnothing(t)} e^{-i2\pi ft} dt \right| 2 \, df}$$

where the $\Delta\varnothing(t)$ represents the phase difference between two signals, the 2B is the frequency band range, set to 1 Hz, $f$ is the frequency, and $T$ is the observation time interval.

The PLM was performed for segments longer than 4 s. By computing the PLM for each couple of brain regions, we obtained a $90 \times 90$ weighted adjacency matrix for each time series and for each subject, in all frequency bands: delta (0.5–4 Hz), theta (4.0–8.0 Hz), alpha (8.0–13.0 Hz), beta (13.0–30.0 Hz), and gamma (30.0–48.0 Hz). Each weighted adjacency matrix was used to reconstruct a brain network (Stam, 2014), where the 90 areas of the AAL atlas are represented as nodes, and the PLM values form the weighted edges. For each trial longer than 4s, and for each frequency band, through Kruskal's algorithm (Kruskal, 1956), the MST was calculated. The MST is a loop-less graph with N nodes and M = N−1 links. The MST was computed to be able to compare topological properties in an unbiased manned (Stam, 2014; Tewarie et al., 2015).

## 2.8 | Graph analysis

Global and nodal (regional) parameters were calculated. In order to characterize the global topological organization of the brain networks, four topological parameters were calculated. The *leaf fraction* (Lf) (Boersma et al., 2013), defined as the fraction of nodes with a degree of 1, provides an indication of the integration of the network, with high leaf fraction conveying a more integrated network. The *degree divergence* (K) (Boersma et al., 2013), a measure of the broadness of the degree distribution, is related to the resilience against targeted attacks. The *tree hierarchy* (Th) (Boersma et al., 2013) is defined as the number of leaf over the maximum betweenness centrality (BC), and is meant to capture the optimal trade-off between network integration and resiliency to hub failure. Finally, the *diameter* (Boersma et al., 2013) is defined as the longest shortest path of an MST, and represent a measure of ease of communication flow across a network. To examine the relative importance of specific brain areas in the brain network, two centrality parameters were calculated: the *degree* (Tewarie et al., 2015), defined as the number of edges incident on a given node, and the BC (Tewarie et al., 2015), defined as the number of the shortest paths passing through a given node over the total of the shortest paths of the network. Before moving to the statistical analysis, all the metrics were averaged across epochs to obtain one value for subject. A pipeline of the processing MEG data is illustrated in Figure 1.

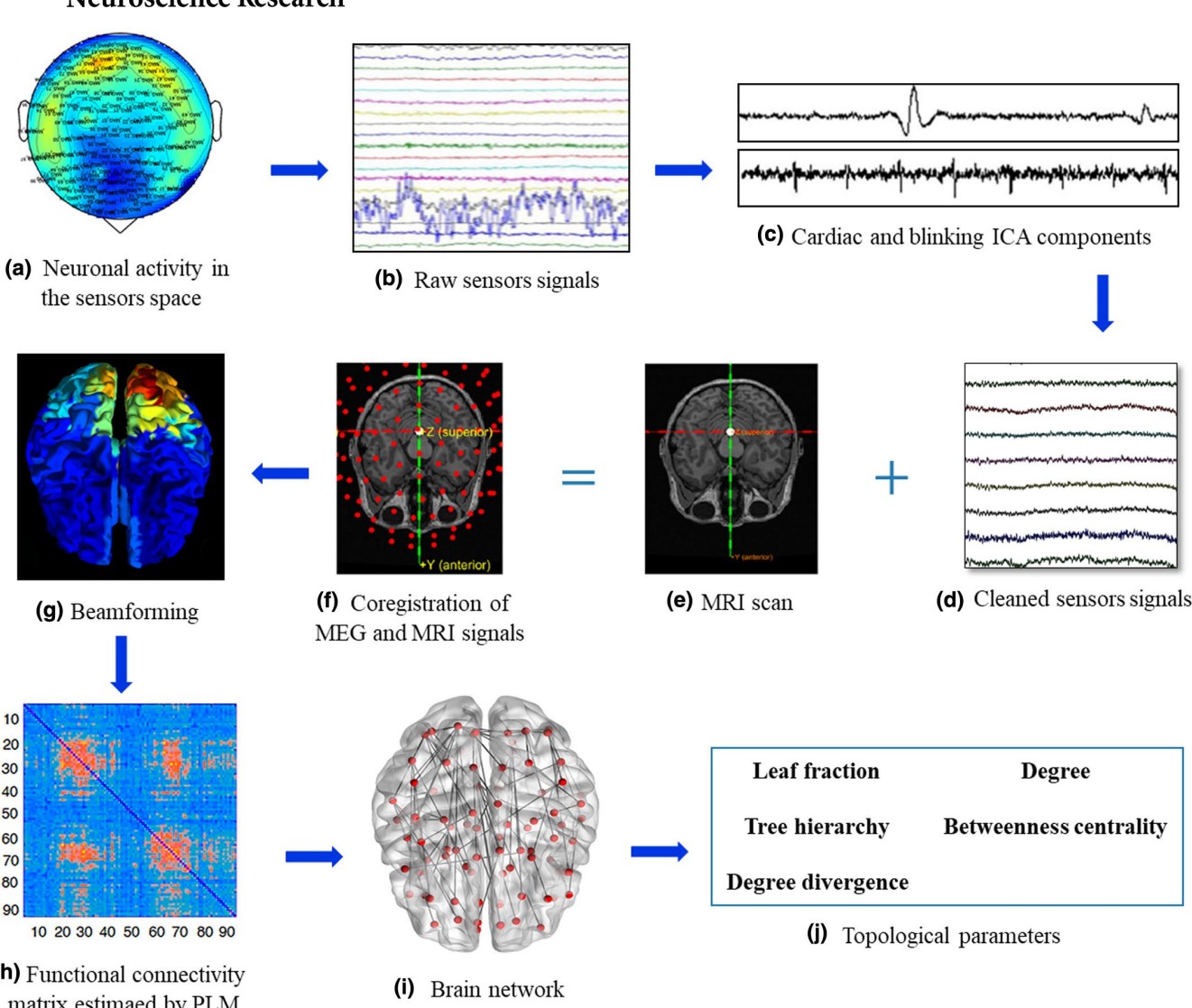

**FIGURE 1** Data analysis pipeline. (a) Neuronal activity recorded in the sensors space using a magnetoencephalography (MEG). Activity in the Alpha frequency (8–13 Hz) is represented. (b) Raw MEG signals recorded by 163 sensors (a subset is shown here). (c) Cardiac (upper) and blinking (lower) artifacts as estimated by the independent component analysis (ICA). (d) Sensors signals after pre-processing and cleaning. (e) Structural magnetic resonance image (MRI) of a subject. (f) The structural MRI, the MEG sensors and the head of the subject are co-registered in the same coordinate system. (g) Through a Beamformer algorithm, source time series are estimated in regions of interest within the brain, according to a parcellation based on the ALL atlas. (h) Functional connectivity matrix estimated using the phase linearity measurement (PLM) which is calculated between each pair of 90 brain regions. In the matrix, rows and columns are the regions of interest, while the entries are the estimated values of the PLM. (i) Brain topology representation based on the minimum spanning tree (MST) reconstruction, where brain regions are represented by red dots and edge are represented as lines. (j) Once a frequency-specific MST has been obtained, it is characterized by topological parameters [Color figure can be viewed at wileyonlinelibrary.com]

## 2.9 | MRI acquisition

MRI images of 24 participants were acquired on a 1.5-T Signa Explorer scanner equipped with an eight-channel parallel head coil (GE Healthcare, Milwaukee, WI, USA). In particular, three-dimensional T1-weighted images (gradient-echo sequence Inversion Recovery prepared Fast Spoiled Gradient Recalled-echo, time repetition = 8.216 ms, TI = 450 ms, TE = 3.08 ms, flip angle = 12, voxel size = $1 \times 1 \times 1.2$ mm$^3$; matrix = $256 \times 256$) were acquired. From the total of 24 women recruited, 22 performed the RM while two subjects refused it and a standard template was used for source reconstruction.

## 2.10 | Psychological evaluation

The psychological assessments were carried out at each of the three phases of the MC. In particular to quantify the self-esteem level, the Rosenberg self-esteem scale (Prezza et al., 1997; Rosenberg, 2015) was used. Additionally the Ryff's test was administered to examine the six dimensions of well-being (autonomy, environmental mastery, personal growth, positive relations with others, purpose in life, and self-acceptance) (Ruini et al., 2003). Finally, in addition to BAI (Beck & Steer, 1990) and BDI (Beck et al., 1996) tests administered at the first experimental session

(as inclusion/exclusion criteria), the tests were re-administered at each time point to exclude the appearance of depressive/anxious symptoms.

## 2.11 | Statistical analysis

Statistical analysis was performed using MATLAB (Mathworks®, version R2013a). The normal distribution of variables was checked with the Shapiro–Wilk test. In order to compare the hormone blood levels at three time points (T1–T3) of the MC, an analysis of variance (ANOVA) for each hormone (estradiol, FSH, LH, progesterone) was performed, followed by post hoc analysis between MC time points, using a paired sample $t$ test. Furthermore, to compare, in all frequency bands, the topological data among the three time points of the MC, we used the Friedman test. All the $p$ values were corrected for multiple comparisons using the false discovery rate (FDR) (Benjamini & Hochberg, 1995) across parameters for each frequency band. Subsequently, the post hoc analysis was carried out using Wilcoxon test. The statistical significance was defined as $p < 0.05$.

If a topological parameter was statistically different in a time point of the MC (as compared to the other time points), we went on to check if its variation across the time points was proportional both to the hormonal levels and to psychological scoring variations. To do this, we calculated the delta (Δ) values, expressed as the variations of the values of each parameter between two time points. Specifically, we calculated both the Δ value between the peri-ovulatory and early follicular time points (Δ T2–T1, defined as the follicular phase), and the Δ value between the mid-luteal and peri-ovulatory time points (Δ T3–T2, defined as the luteal phase), for the topological parameters, hormonal blood levels (estradiol, FSH, LH, and progesterone) and psychological scoring (Rosenberg and Ryff tests).

To test the hypothesis that during the MC, the topological changes were related to hormonal fluctuations, we built a linear model to predict the topological values based on sex hormone blood levels. Specifically, we considered the variations in the topological parameters (Δ T2–T1 and Δ T2–T3) as the dependent variable, while the estradiol, progesterone, LH, and FSH variations (Δ T2–T1 and Δ T3–T2) were used as predictors. Moreover, nuisance variables were added to account for the presence of repeated measures, as well as for age, education, and MC length. To make the prediction of our model more reliable and to test its generalization capacity, we used a leave-one-out cross-validation (LOOCV) technique. Expressly, we built $n$ multilinear models (where $n$ is the size of the sample included in the model), excluding each time a different element from the model, and verifying the ability of the model to predict the topological value of the excluded element.

Based on the results of the multilinear model, we performed a correlation analysis using the Spearman's correlation test, in order to investigate the relationship among topological changes, hormone variation, and changes of the psychological tests scores (self-esteem

and well-being with the six relative dimensions) during the follicular (Δ T2–T1) and luteal (Δ T3–T2) phases of the MC. Finally, to explore the possibility that the correlations among topological changes, hormone variation, and the changes of the psychological tests scores could be driven by multicollinearity, we estimated the variance inflation factor (VIF) (Belsley et al., 2005; Snee, 1983). All $p$ values were corrected for multiple comparisons using FDR and a (corrected) $p$ value <0.05 was accepted as significant.

## 3 | RESULTS

Preliminarily, we checked the differences in blood concentration of the hormones (LH, FSH, progesterone, and estradiol) at the three times points of the MC. The ANOVA showed significant differences for each hormone: LH ($F(2.69) = 16.28$, $p < 0.001$, $p_{FDR} < 0.001$), FSH ($F(2.69) = 25.39$, $p < 0.001$, $p_{FDR} < 0.001$), progesterone ($F(2.69) = 83.2$, $p < 0.001$, $p_{FDR} < 0.001$), estradiol ($F(2.69) = 27.64$, $p < 0.001$, $p_{FDR} < 0.001$). Post hoc analysis showed that each hormone blood level was statistically different among all time points was statistically different from each other, in each hormone, except for LH in T1 versus T3 and FSH in T1 versus T2 (Table 2).

### 3.1 | Analysis of the topological parameters

The nodal analysis showed significant difference in the BC of the right posterior cingulate gyrus (rPCG) ($\chi^2$ ($df = 2$, $N = 24$) = 15.2500, $p = 4.8 \times 10^{-4}$, $p_{FDR} = 0.043$), in the alpha band. In detail, the post hoc analysis showed significantly higher BC in the rPCG during the peri-ovulatory phase, as compared to the early follicular ($p = 0.0003$) and mid-luteal ($p = 0.0055$) phases (Figure 2).

With regard to the global topological parameters, the Lf ($\chi^2$ ($df = 2$, $N = 24$) = 10.7500, $p = 0.0046$, $p_{FDR} = 0.009$) and the Th ($\chi^2$ ($df = 2$, $N = 24$) = 12.3333, $p = 0.0021$, $p_{FDR} = 0.008$) were reduced in the alpha band. More specifically, the post hoc analysis revealed a reduction in the network integration in the peri-ovulatory phase as compared to both the early follicular (Lf $p = 0.016$; Th $p = 0.032$) and the mid-luteal (Lf $p = 0.004$; Th $p = 0.006$) phases (Figure 2). No statistically significant difference was found, after FDR correction, in any other nodal and global parameter, nor in any other frequency band. The topological parameters that showed significant variations during the MC (the BC in the rPCG, the Lf, and the Th) became parameter of interest for follow-up linear model analyses and correlations.

### 3.2 | Multilinear model analysis

The relationship between hormonal level variations and topological parameters was investigated through a linear model that predicts the BC variance of the rPCG, Lf, and Th (Figure 3). We found that

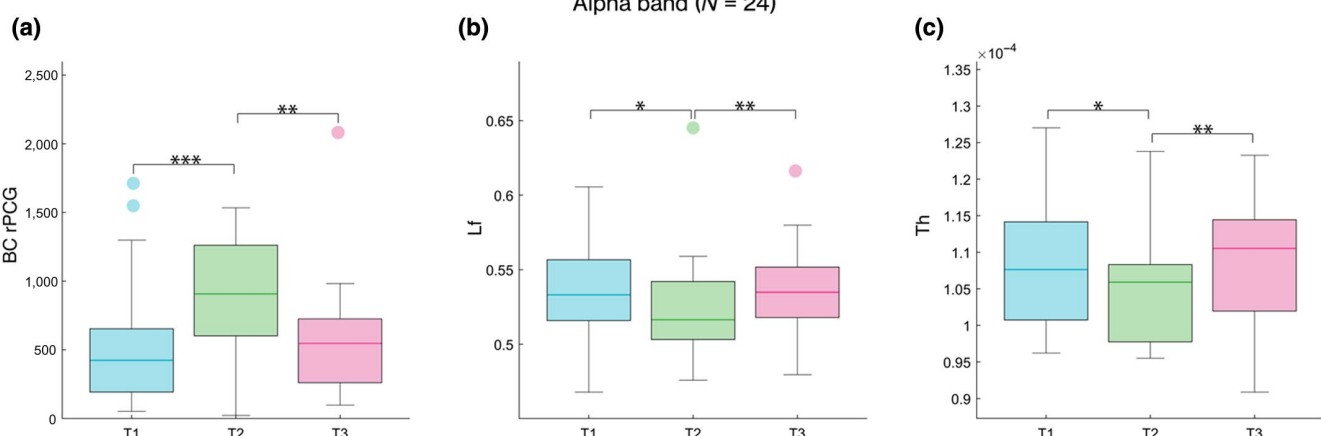

**FIGURE 2** Brain topology comparison. The box plots refer to the comparison of topological parameters in alpha band, estimated in 24 women (N = 24) during the MC. From left to right, (a) the BC of the right posterior cingulate gyrus (rPCG), (b) the leaf fraction (Lf), and (c) the tree hierarchy (Th), respectively. In each box plot, the values are shown at early follicular (T1), peri-ovulatory (T2) and mid-luteal (T3) phase. The upper and lower bound of the rectangles refer to the 25th to 75th percentiles, the median value is represented by a horizontal line inside each box, the whiskers extend to the 10th and 90th percentiles, and further data are considered as outliers and represented by the filled circles. From left to right, the box plots show the significantly higher BC in the rPCG during the peri-ovulatory phase, as compared to the early follicular (p = 0.0003) and mid-luteal (p = 0.0055) phases, and the reduction in the network integration in peri-ovulatory phase as compared to both the early follicular (Lf p = 0.016; Th p = 0.032) and the mid-luteal (Lf p = 0.004; Th p = 0.006) phases. Significance p value: *p < 0.05, **p < 0.01, ***p < 0.001 [Color figure can be viewed at wileyonlinelibrary.com]

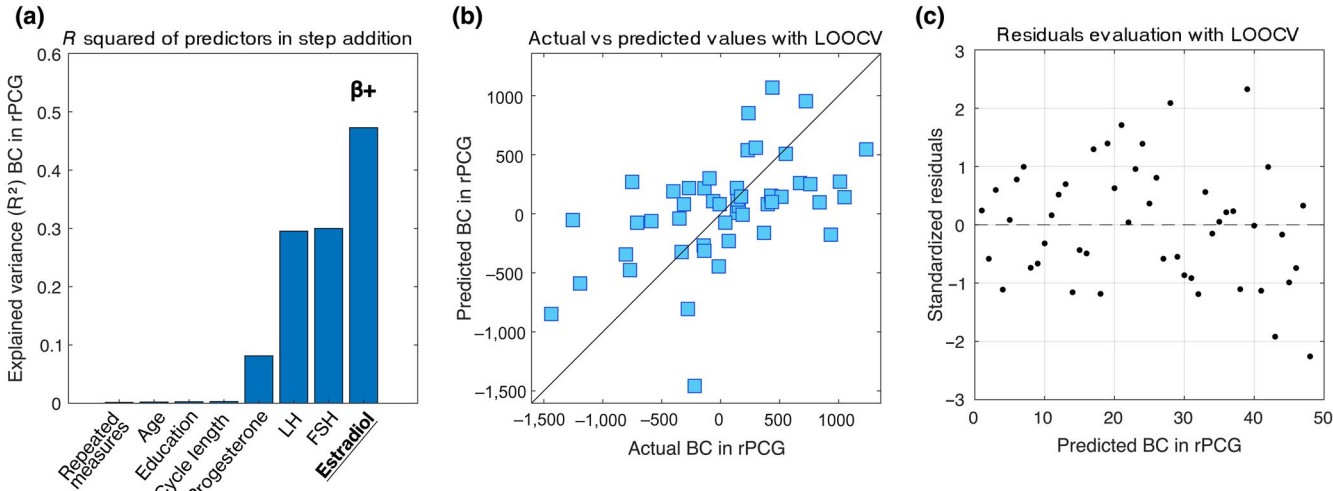

**FIGURE 3** Multilinear model with leave-one-out cross-validation (LOOCV). The model aims to predict the topological variation of the brain network expressed by the BC changes during the MC (Δ T2–T1 and Δ T3–T2) of the right posterior cingulate gyrus (rPCG). (a) Explained variance of the additive model composed of four nuisance variables (repeated measures, age, education, cycle length), and four predictors (progesterone, luteinizing hormone (LH), follicle-stimulating hormone (FSH), estradiol). Significant predictor in underlined text; positive coefficient indicated with β+. (b) Scatter plot of the Observed topological values versus the topological values predicted by the model with LOOCV. (c) Scatter plot of the standardized residuals (standardization of the difference between observed and predicted (LOOCV) values). The distribution results symmetrical with respect to the 0, with a standard deviation lower than 2.5 [Color figure can be viewed at wileyonlinelibrary.com]

the model yielded significant predictions of the BC of the rPCG ($R^2$ = 0.47), with estradiol being a significant predictor for the model (p < 0.001), with positive beta coefficients. The prediction of the model and the distribution of the residuals are shown in Figure 3 (panels B and C). The same model was applied to global topological parameters, but no significant results were obtained.

## 3.3 | Topological brain network parameters and hormone blood levels

Correlation analysis between the variations of the brain network topological parameters and the concurrent variations in the hormonal blood levels, showed a statistically significant direct correlation between the

Δ T3–T2 values of the BC of the rPCG in the alpha band and those of estradiol ($r = 0.541$, $p = 0.007$, $p_{FDR} = 0.029$) (Figure 4). A correlation between the Δ T2–T1 values of the BC of the rPCG and those of the estradiol was observed only before FDR correction ($r = 0.404$, $p = 0.027$, $p_{FDR} = 0.453$). No correlation was demonstrated between the global topological parameters and the other hormonal levels.

## 3.4 | Topological brain network parameters and psychological scores

Correlation analysis between the brain network parameters and the psychological scores (Figure 5) showed a significant direct correlation between the Δ T3–T2 values (but not Δ T2–T1) of the BC of the rPCG and those of a dimension of the well-being test, namely the environmental mastery ($r = 0.534$, $p = 0.007$, $p_{FDR} = 0.043$). No correlation was demonstrated between the variations (Δ T2–T1 and Δ T3–T2) of individual well-being dimensions and the global topological parameters. Regarding the self-esteem, no correlation was found between the questionnaire scores and both the nodal and global topological parameters.

## 3.5 | Hormone blood levels and psychological scores

Correlation analysis between the Δ T3–T2 values of the hormonal levels and those of the psychological scores showed a statistically significant correlation between estradiol and environmental mastery

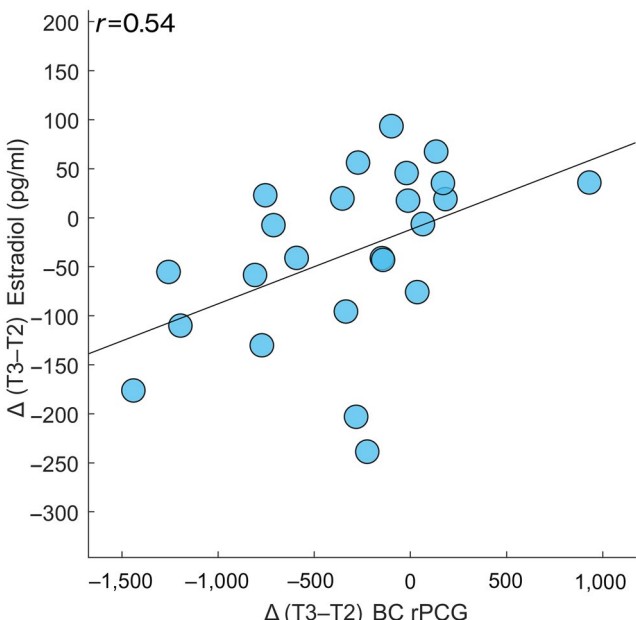

**FIGURE 4** Correlation between topological data and hormone blood levels. Spearman's correlation between the Δ values (here expressed as the difference between the mid-luteal (T3) and peri-ovulatory (T2) phases of the MC) of the BC of the right posterior cingulate gyrus (rPCG) and the Δ values of the estradiol levels along the MC ($p = 0.007$, $p_{FDR} = 0.028$) [Color figure can be viewed at wileyonlinelibrary.com]

($r = 0.712$, $p < 0.001$, $p_{FDR} < 0.001$) (Figure 6) while the analysis of the Δ T2–T1 failed to show any correlation between estradiol (or any other hormone) and any dimensions of the Ryff's test. No correlation was found between the variations (Δ T2–T1 and Δ T3–T2) of hormonal levels and self-esteem questionnaire scores.

Finally, we checked if the relationship between the BC of the rPCG, the estradiol levels and the environmental mastery scores could be driven by multicollinearity. The VIF (Belsley et al., 2005; Snee, 1983) confirmed that no multicollinearity was present among those three elements (VIF values: BC rPCG = 1.31, estradiol = 1.95, environmental mastery = 1.91).

## 4 | DISCUSSION

In the present study, we set out to test the hypothesis that sex hormone changes, as they occur across the MC, may affect the topological configuration of brain networks, as well as modulate the frequently observed mood changes (Halbreich et al., 2007; Yonkers & Simoni, 2018). We showed that during the MC the topological features of the brain network undergo profound rearrangements under the effect of sex hormones, as highlighted by changes in both nodal and global topological parameters. In particular, we showed in the alpha band, during the peri-ovulatory phase, increased BC in the rPCG and reduced Lf and Th, as compared to both the early follicular and mid-luteal phases. Through a multilinear model, we demonstrated that nearly 50% of the variance of the changes of the BC could be explained by the estradiol levels. The correlation analysis evidenced that the BC values increase in the rPCG was positively correlated with the changes in the blood levels of estradiol during the luteal phase (during the follicular phase a statistical significance was observed before FDR correction). We also demonstrated that the increase in the BC was positively correlated with the environmental mastery (one of the six dimensions of the well-being test) that, in turn, during the luteal phase was correlated with the estradiol levels.

The PCG is described as "an enigmatic cortical region" (Leech & Smallwood, 2019). If, on the one hand, the high metabolic expenditure and the number of cortical and subcortical connections point at the PCG as structural and functional hub, on the other hand, growing evidence shows that the PCG tends to deactivate in response to attention demanding tasks (Raichle et al., 2001). Accordingly, the PCG displays increased activity when the subject is involved in internally directed task such as retrieving autobiographical memories, planning for the future or wandering freely with the mind (Addis et al., 2007; Gusnard et al., 2001; Mason et al., 2007). Recent studies suggest that the PCG may play a crucial role in the stepwise mechanisms of integration of specialized perceptive processes (i.e., visual, auditory, or sensory) into higher levels of abstraction. Other works suggest that the PCG may play a role in assessing the significance of decision outcome, being important in balancing between risk-prone and risk-adverse behaviors (Leech & Smallwood, 2019).

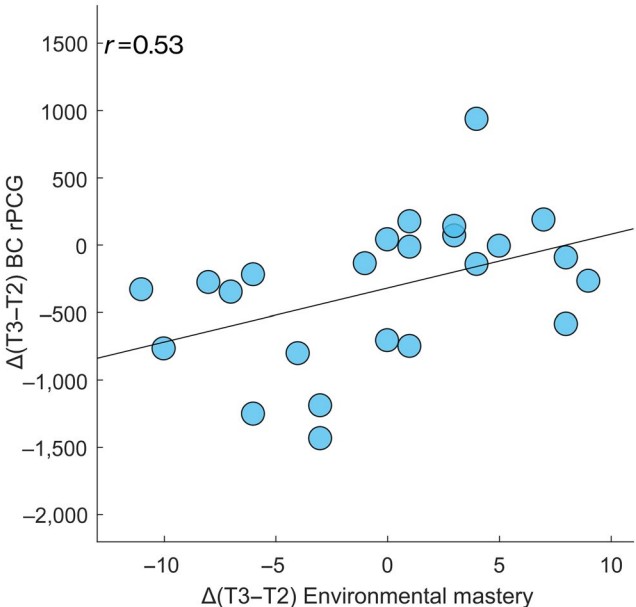

**FIGURE 5** Correlation between topological data and psychological dimensions of well-being test. Spearman's correlation between the Δ values (here expressed as the difference between the mid-luteal (T3) and peri-ovulatory (T2) phases of the MC) of the BC of the right posterior cingulate gyrus (rPCG) and the Δ values of the psychological dimension of well-being test (Environmental mastery scores) along the MC ($p = 0.007$, $p_{FDR} = 0.043$) [Color figure can be viewed at wileyonlinelibrary.com]

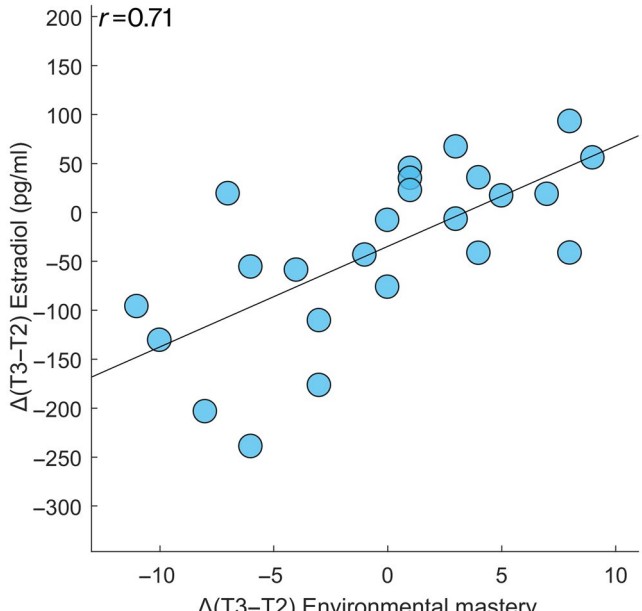

**FIGURE 6** Correlation between hormone blood levels and psychological dimensions of well-being test. Spearman's correlation between the Δ values (here expressed as the difference between the mid-luteal (T3) and peri-ovulatory (T2) phases of the MC) of estradiol and the Δ values of the psychological dimension of well-being test (Environmental mastery scores) along the MC ($p < 0.001$, $p_{FDR} < 0.001$) [Color figure can be viewed at wileyonlinelibrary.com]

It is noteworthy that the PCG change is not symmetric. This fact may be associated with a different influence of the sex hormones on the right and left PCG, possibly modulating the expression of affective behavioral styles. Hwang et al. (2008), in a MEG study, demonstrated an asymmetry on the way the brain is modulated by sex hormones during the MC. In particular, higher right frontal activity was observed during the ovulation phase and a higher left activity during the menstruation phase. However, they found this left–right asymmetry in the frontal regions of the brain, while our data points at the posterior brain regions. Nonetheless, it is interesting to note that the DMN areas possess long-distance projections to the anterior cingulate areas via the PCG (Baker et al., 2018). Furthermore, it has been shown that the asymmetry at rest between the right and left sides of the brain represents a reliable measure of individual affective style (Sutton & Davidson, 1997). In particular, greater alpha activity in the right regions corresponds to a personality trait sensitive to negative affective stimuli, while greater alpha activity in the left corresponds to a personality trait sensitive to positive affective stimuli. Regarding the possible mechanisms of action of sex hormones in determining cortical asymmetry, the hypothesis of progesterone-mediated hemispheric decoupling should be mentioned (Hausmann & Güntürkün, 2000; Hodgetts & Hausmann, 2018). The authors suggested that high levels of progesterone during the luteal phase lead to a functional decoupling of the two hemispheres and this, in turn, to reduced lateralization. In contrast, Weis et al. reported that estradiol alone, not progesterone, was associated with a reduced interhemispheric inhibition and cortical asymmetry (Weis et al., 2008).

Furthermore, we showed a statistically significant reduction of the Lf and the Th in the alpha band during the peri-ovulatory phase, as compared to the early follicular and mid-luteal phases. These data might suggest a shift toward a less centralized organization of the brain network (Boersma et al., 2013) in which the information flow is less reliant on any single node, with consequent improved resiliency to targeted attacks (Stam, 2014; Tewarie et al., 2015). These results could be summarized as a better global efficiency which is expression of an optimal organization of the brain network during the peri-ovulatory phase, in terms of an optimal trade-off between efficient communication and resiliency (Sorrentino et al., 2018). However, further studies are needed addressed specifically to this topic.

The multilinear model demonstrated that there is a relationship between the topological variation and the hormonal fluctuations that occur during the MC, in fact nearly 50% of the variance of the changes of the BC in the rPCG during the MC can be explained by the changes in estradiol blood levels. Multiple works have tried to disentangle hormone-specific influences on the brain networks. However, the literature is largely inconsistent, even when limiting oneself to the effects of hormones on brain connectivity alone. Several studies have shown the involvement of the estradiol in both the structure and the function of the brain. In particular, it has been observed that estradiol affects the activity of the right anterior hemisphere (Hwang et al., 2008). Pletzer et al. (2019) demonstrated that the left hippocampus is highly activated during the pre-ovulatory phase, while its activation drops during the luteal phase, suggesting that estradiol and progesterone have opposite effects on the hippocampus.

Furthermore, MRI studies have reported increased gray matter volumes in the hippocampus during the pre-ovulatory phases (Pletzer et al., 2019; Protopopescu et al., 2008). A resting state MRI study found a significant positive correlation between progesterone (but not estradiol) and the EC in the dorsolateral prefrontal cortex in a single woman scanned 32 times across four MCs (Arélin et al., 2015). However, other studies did not find any correlation between resting state activity and neither progesterone nor estradiol (Petersen et al., 2014). Very recently, Pritschet et al. (2020), in a very elegant study, demonstrated the crucial effect of estradiol on brain network. The authors performed a dense-sampling protocol, scanning the same woman for 30 consecutive days. One year later the same woman repeated the protocol while she was under hormonal therapy, as to selectively suppress progesterone synthesis, while leaving estradiol unaffected. In the second experimental setting, the authors were able to confirm the previous results. Finally, Franke et al., through a machine learning approach could accurately classify the cycle phase, highlighting the relationship between estradiol and morphological brain characteristics (Franke et al., 2015).

Our observation of the positive correlation between the increase in BC, suggesting a greater topological centrality of the rPCG within the cerebral network, and higher levels of estradiol, does not find an immediate and unambiguous explanation. Albeit within a purely speculative perspective, we notice that the greater centrality of the PCG is coupled to the levels of estradiol, showing that the role of this region within the network is more prominent during the moment of fertility. Observing this phenomenon from an evolutionistic perspective, one could think that, when fertility is at its peak a quick and effective evaluation of the relative risks and rewards associated with the potential mate would be adaptive. The PCG might have implications in the top-down control in decision-making as in the choice of the partner (Penton-Voak & Chen, 2004; Roney et al., 2006).

Some studies are in line with this otherwise speculative hypothesis (Gangestad & Simpson, 2000; Jones et al., 2008; Pastor et al., 2008; Pawlowski & Jasienska, 2005; Penton-Voak & Perrett, 2000; Proverbio et al., 2008, 2011; Roney & Simmons, 2008; Rozenkrants et al., 2008). For example, Proverbio et al., in an event-related potential, source reconstructed EEG study revealed strongest activations in the PCG when presenting scenes in which two people performed "affective" actions, while the superior temporal sulcus, an area included in the mirror neuron system, was activated by cooperative scenes (Proverbio et al., 2011). Furthermore, this observation seems to be gender specific, since women show improved comprehension of unattended social scenes as compared to men. Rupp et al. (2009) used fMRI to measure brain activity in 12 heterosexual women as they evaluated pictures of masculinized or feminized male faces, during both the follicular and luteal phase. They found that the brain regions involved in face perception, decision-making and reward processing, including the PCG, responded more strongly to masculinized faces as compared to feminized ones. Interestingly, the authors showed that such process was influenced by the estradiol hormonal levels. Further investigation is prompted to understand if

the observed behavioral changes are specifically related to mating or, rather, a more aspecific increase toward social engagement.

An experience shared by a very large number of women of childbearing age is an emotional lability during the luteal phase, in the days immediately before the menstruation (Campagne & Campagne, 2007). This condition can take on clinical relevance in the form of PMS or even grow to a dysphoric clinical picture as in the case of PMDD (Parker & Brotchie, 2010; Payne et al., 2009). A number of studies have investigated the role of sex hormones in PMS/PMDD, but no abnormal levels have been established (Dubol et al., 2020; Rubinow & Schmidt, 1992) although with inconsistency (Nevatte & O'Brien, 2013). At moment, the hypothesis with the stronger consensus claims a maladaptive response of the brain regions involved in affective processes to the physiological fluctuations of the sex hormones (Comasco & Sundström-Poromaa, 2015). In this study, we sought to provide evidence about the possible correlation between clinically under-threshold affective modifications and both topological changes and sex hormone fluctuations observed along the MC. We showed a positive correlation during the luteal phase between the BC values of rPCG and the environmental mastery of the Ryff's test which assess the ability "to choose or create contexts suitable to personal needs and value" (Ryff, 1989). This result is in agreement with Villani et al. that found that the environmental mastery is the only unstable domain of the Ryff's test during the MC (Villani et al., 2017). Furthermore, we showed a correlation between the environmental mastery dimension and estradiol levels. Our data demonstrate that during the peri-ovulatory time point, when estradiol reaches its peak, the BC values of the rPCG peak as well. At same time, the condition of well-being, as sense of competence in managing the environment and control the external activities, correlates positively with both the BC of the rPCG and estradiol blood levels. The combination of these observations suggests that the sex hormones interfere with the sense of well-being, possibly by changing the topological features of the rPCG, a brain region specifically involved in the top-down computation of emotional stimuli (Leech & Smallwood, 2019). It is worthy of note that these correlations were found only during the luteal phase of the cycle (defined as the difference between mid-luteal and peri-ovulatory phases of MC), when the estradiol levels drop. This could be in agreement with the well-established prevalence of psychopathological symptoms in the premenstrual period, when a sudden reduction of the estradiol blood levels is observed (Dubol et al., 2020; Halbreich, 2003; Schmidt et al., 1998). Finally, our results provide relevant information for all the studies that use brain imaging to compare multiple groups. In fact, our work suggests that the brain functioning may be influenced by the hormonal profile in women, and this information should be taken into consideration to avoid a biased comparison.

Some limitations of our study deserve attention. First, although the presence of depression and/or anxiety was ruled out using the BDI and the BAI, the lack of premenstrual symptoms was verified only

through targeted clinical investigation by an experienced gynecologist, and not with structured questionnaires. Secondly, we estimated the day of ovulation according to backward counting, and we checked hormone levels during the peri-ovulatory phase. However, a single hormone measurement may not be sufficient to accurately determine the phase of the cycle (Becker et al., 2005; Sundström Poromaa & Gingnell, 2014). To be more accurate, one should perform MEG recordings, hormone assays, and psychological assessment daily, and then define the registration corresponding to the peak of LH, but this would not be feasible. Third, it should be considered that the sample was drawn from a population with high socio-cultural level, young age, and consisted almost entirely of nulliparous. Finally, the source reconstruction of two subjects was performed by using a standard template, which might yield less accurate results.

## 5 | CONCLUSIONS

In conclusion, our exploratory study shows the pivotal role of sex hormones on large-scale functional organization of the brain as well as on the possible relationship with the psychological state across the MC. Furthermore, our work may have widespread implications for all clinical neuroimaging studies, given that the comparison of topological parameters between groups should account for the expected physiological variations occurring in women as function of the phase of the MC.

## DECLARATION OF TRANSPARENCY

The authors, reviewers and editors affirm that in accordance to the policies set by the *Journal of Neuroscience Research*, this manuscript presents an accurate and transparent account of the study being reported and that all critical details describing the methods and results are present.

### ACKNOWLEDGMENTS
We thank all the women who contributed to the realization of this study, Professor Laura Mandolesi for her precious advice, and the two anonymous Reviewers for the many valuable inputs.

### COMPETING INTERESTS
The authors declare that there is no conflict of interest regarding the publication of this paper.

### AUTHOR CONTRIBUTIONS
All authors had full access to all the data in the study and take responsibility for the integrity of the data and the accuracy of the data analysis. *Conceptualization*, M.L. and E.T.L.; *Methodology*, E.T.L. and P.S.; *Investigation*, M.L., E.T.L., P.S., L.S., R.R., R.M., and M.P.; *Formal Analysis*, M.L., E.T.L., P.S., R.R., and G.P.; *Resources*, G.S. and P.F.; *Writing – Original Draft*, M.L., E.T.L., P.S., G.S., L.S.; *Writing –Review & Editing*, F.L.; *Visualization*, M.L. and E.T.L.; *Project Administration*, G.S.; *Funding Acquisition*, G.S.

## PEER REVIEW
The peer review history for this article is available at https://publons.com/publon/10.1002/jnr.24898.

## DATA AVAILABILITY STATEMENT
The data used to support the findings of this study are available from the corresponding author upon reasonable request, and provided authorization by the local ethical committee.

## ORCID
*Marianna Liparoti* https://orcid.org/0000-0003-2192-6841
*Emahnuel Troisi Lopez* https://orcid.org/0000-0002-0220-2672
*Laura Sarno* https://orcid.org/0000-0002-5578-2885
*Rosaria Rucco* https://orcid.org/0000-0003-0943-131X
*Roberta Minino* https://orcid.org/0000-0002-8416-0807
*Matteo Pesoli* https://orcid.org/0000-0002-9176-6962
*Giuseppe Perruolo* https://orcid.org/0000-0003-0479-8729
*Pietro Formisano* https://orcid.org/0000-0001-7020-6870
*Fabio Lucidi* https://orcid.org/0000-0003-2203-9566
*Giuseppe Sorrentino* https://orcid.org/0000-0003-0800-2433
*Pierpaolo Sorrentino* https://orcid.org/0000-0002-9556-9800

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

## SUPPORTING INFORMATION

Additional Supporting Information may be found online in the Supporting Information section.

Transparent Peer Review Report

Transparent Science Questionnaire for Authors

