## [Transparent Peer Review Report · Journal of Neuroscience Research]

Functional brain network topology across the menstrual cycle is estradiol dependent and correlates with individual well-being

Liparoti^A, Marianna; Troisi Lopez, Emahnel; Sarno, Laura; Rucco, Rosaria; Minino, Roberta; Pesoli, Matteo; Perruolo, Giuseppe; Formisano, Pietro; Lucidi, Fabio; Sorrentino, Giuseppe; Sorrentino, Pierpaolo

Review timeline:

Submission date: 10 November 2020
Editorial Decision: Major Modification (17 February 2021)
Revision Received: 7 April 2021
Editorial Decision: Minor Modification (4 May 2021)
Revision Received: 11 May 2021
Accepted: 15 May 2021

Editor 1: Inger Sundstrom Poromaa
Editor 2: Junie Warrington
Reviewer 1: Rachel Zsido
Reviewer 2: Esmeralda Hildago-Lopez

1st Editorial Decision

Decision letter

Dear Professor Sorrentino:

Thank you for submitting your manuscript to the Journal of Neuroscience Research. We have now received the reviewer feedback and have appended those reviews below. As you will see, the reviewers find the question addressed to be of potential interest. Yet, they do not find the manuscript suitable for publication in its current form.

If you feel that you can adequately address the concerns of the reviewers, you may revise and resubmit your paper within 90 days. It will require further review. Please explain in your cover letter how you have changed the present version and submit a point by point response to the editors' and reviewers' comments. If you require longer than 90 days to make the revisions, please contact Dr Junie Warrington (jpwarrington@umc.edu). To submit your revised manuscript: Log in by clicking on the link below <https://rex-prod.resxchange.com/submissionBoard/1/84177b9b-0d08-4584-9ce5-4b6fa5472dc5/current>

(If the above link space is blank, it is because you submitted your original manuscript through our old submission site. Therefore, to return your revision, please go to our new submission site here (submission.wiley.com/jnr) and submit your revision as a new manuscript; answer yes to the question "Are you returning a revision for a manuscript originally submitted to our former submission site (ScholarOne Manuscripts)? If you indicate yes, please enter your original manuscript's Manuscript ID number in the space below" and including your original submission's Manuscript ID number (jnr-2020-Nov-9204) where indicated. This will help us to link your revision to your original submission.)

Thank you again for your submission to the Journal of Neuroscience Research; we look forward to reading your revised manuscript.

Best Wishes,

Professor Inger Sundström Poromaa
Associate Editor, Journal of Neuroscience Research

Dr Junie Warrington
Editor-in-Chief, Journal of Neuroscience Research

Associate Editor: Sundström Poromaa, Inger
Comments to the Author:
(There are no comments.)

Reviewer: 1

Comments to the Author

Regarding the potential need for additional analyses / statistical analysis concerns - please see point 2 and point 24.

The authors use MEG to investigate how functional brain network topology changes across three important menstrual cycle time-points in 24 healthy female participants. Additionally, they measured levels of sex hormones and administered psychological questionnaires at each assessment in order to relate measures of the brain, sex hormones, and psychological well-being of the participants.

I found this to be an important study with novel findings, particularly as the authors consider how well-being and self-esteem can change across the menstrual cycle even at subclinical levels. A major strength of the study is the estimation of individual menstrual cycles through the use of transvaginal pelvic ultrasonography and measuring sex hormones in the participants' blood, as well as the use of graph theory approaches to characterize global and nodal brain topological organization. Before supporting acceptance, however, several comments would have to be addressed.

Major concerns:

1. Why were the questionnaires Rosenberg Self-Esteem Scale and the Ryff's Test chosen, especially the former (as self-esteem is not mentioned once in the Introduction)? It would be helpful if the Introduction reviewed more previous literature related to specific changes in self-esteem and emotional well-being across the menstrual cycle. The Introduction does explain that sex hormones influence brain function, but lacks a review of specific changes in emotion/mood across the menstrual cycle. PMDD is briefly discussed and then the authors state that the knowledge on menstrual cycle-related changes in emotional experiences (below clinical threshold) are "sparse and anecdotal" -- but there is existing literature that should be discussed. If space is limited, I would suggest condensing some of the text from the second paragraph (on how sex hormones influence gender differentiation and physiological brain changes across the lifespan); or from the final paragraph of the Introduction, in which there is a lot of explanation of methods that may be too detailed for the Introduction. Both of the paragraphs could be condensed.

2. I apologize if I have missed something. Is it standard to include all data-points (all 24 subjects, two time-points) into a single correlation analysis (48 data-points, such as is done in results section 3.2, 3.3, 3.4)? I believe Spearman's correlation assumes that all observations are independent from one another. This makes it a little more difficult to interpret. Why don't the authors investigate these correlations by menstrual cycle phase?

3. The Discussion section would benefit from having a clear limitations section (the Transparent Science Questionnaire states that this is in the 4th paragraph of page 17 but there is no paragraph dedicated to limitations). For example, the authors discuss the importance of measuring subclinical, subtle changes in mood and emotion related to the menstrual cycle. But the BDI and BAI used are typically assessed over the past week and are not menstrual cycle specific (for example, like the Premenstrual Symptoms Screening Tool, designed to assess subtler changes in mood related to the late luteal phase). The authors could also include any limitations regarding the study sample or methods used.

Minor concerns:

4. The second "the" in the title is not necessary.

5. Abstract: the authors mention in the Abstract (as well as in the main text) that the menstrual cycle is the only sex hormone-related phenomenon that repeats itself cyclically. This wording should be adjusted, as there are other examples of this (e.g., diurnal fluctuations in testosterone in males).

6. Abstract: After reading the discussion, I understood what the authors meant by "reproductive psychology" in the abstract. However, I believe this is too complicated a term/argument to be used without explanation. The authors should either explain what they mean by this in regards to their findings, or remove it from the abstract.

7. Abstract: It would be informative to mention what aspects of well-being that they analyzed/found significant. The methods should mention the psychological questionnaires used in the study. Otherwise, main parts of the study's analyses (for example, the self-esteem questionnaire) go unmentioned.
8. Graphical abstract: Please label all x and y axes. Please state N (perhaps in title, after "Alpha Band"). In the legend, PCG should be rPCG (as in axes). I believe "letter" should be "latter". Without reading the manuscript, the reader does not know what T1, T2, or T3 means. It may be clearer to actually label the axis as follicular, ovulatory, and luteal in the graphical abstract.
9. Significance statement: The text may require another proofread (e.g., "a significantly change"). There are a few other examples throughout the manuscript (nothing that makes it incomprehensible, but things like "on a sample of twenty-four females" and "trough mechanisms").
10. Introduction: The authors hypothesize that brain network topology rearranges across the menstrual cycle as a function of hormone levels, and that these changes are associated with changes in affective condition. If applicable, can the authors be more specific (for example, would they expect a positive/negative correlation between changes in hormones and changes in BC?). I ask because this is a very rich study with a lot of measures (looking at whole brain, all frequency bands, multiple sex hormones, different questionnaires, etc..).
11. Introduction: Citations are required for lines 70-73 ("Hormonal modulation... differentiation") and 111-115 ("In the last... by such areas").
12. Introduction: Just checking if "vegetative" is the correct terminology (line 88)?
13. Introduction: BOLD is never defined, and fMRI and EEG are defined twice.
14. Introduction: Line 139 is missing something: "...with the change negatively correlated to the (?) and estradiol levels, suggesting that the latter..."
15. Introduction: Line 141 - a null finding is not enough justification for something being "poorly understood." If there are mixed findings or limited research on the topic, then this should be stated.
16. Methods: Were there any other exclusion criteria, such as medication use (especially that affects the CNS), tobacco/alcohol use, or positive drug tests?
17. Methods: The back-counting method (line 192) needs to be briefly explained, and a citation provided.
18. Methods: What is the "specific questionnaire" (lines 248-249) for drowsiness?
19. Methods: Section 2.9 begins by stating MR images were acquired from 24 participants. As two participants refused to do the MRI scan, shouldn't this be 22?
20. Methods: As Section 2.10 describes the psychological evaluations, it would be helpful if the 6 subdomains that were analyzed were mentioned in this section.
21. Methods: Apologies that I am not so familiar with MEG recordings, is it standard to have a break within a 7-minute recording (i.e., at the 3.5 minute mark)?
22. Results: For the sake of length, methods descriptions could be removed from the Results section. Each result section currently starts off with 2-4 lines explaining why each analysis was done (which should already be clear from the Methods section).
23. Results: Negative findings should also be reported in the Results section (for example, the self-esteem questionnaire is not mentioned in the entire Results section, LH and FSH are not mentioned in Section 3.4).
24. Results: Given that BC rPCG correlates to estradiol, BC rPCG correlates to environmental mastery, and estradiol relates to environmental mastery, couldn't this be driven by multicollinearity? For example, between BC rPCG and estradiol (in relation to both correlating with environmental mastery).
25. Discussion: The "frequently observed mood changes" should be briefly reviewed (or at least citations

included), particularly regarding the measures of interest in this study (e.g., self-acceptance, environmental mastery).

26. Discussion: Line 447 - please specify 'in the alpha band' when referring to the "significant reduction of the Lf and the Th".

27. Discussion: Line 447-460 - It would be interesting if the authors could discuss the role of sex hormones for this finding. The authors explain that there may be "better global efficiency" during the ovulatory phase when hormones are high - do the authors have any thoughts on how increased levels of estradiol, FSH, and LH may help facilitate this (perhaps from a mechanistic standpoint)?

28. Discussion: Making an evolutionary argument for changes in brain network topology in the ovulatory phase must be done so cautiously. I suggest that the authors make it clearer that this is a highly speculative perspective. The sentence starting on line 499 ("Furthermore..") requires citations. Please correct me if I am mistaken, but I believe the authors are citing Rupp et al., 2009, to suggest that women have evolved to be able to most effectively respond to a potential mate when they are fertile. If this is the case, then it is necessary to mention that that study was conducted in heterosexual women (if the authors are suggesting that male faces are more attractive as mates). If the argument is that this is related to reproduction, I would suggest providing more evidence for this being something specifically related to potential mates/sex/etc. (versus a general increase in affect/social behavior).

29. General comment: While sometimes appropriate, authors should be careful using causal, non-flexible language (e.g., "prove", "provoke", "predict").

30. Table: A table would be helpful (listing participant information such as mean and standard deviation for age, cycle length during participation, hormone values per time-point, etc.).

31. Figure 1: Are the images in Figure 1 just representative or based on the data? While a representation of the pipeline is useful for the reader, I think this figure requires more work - as there are no labels nor text anywhere within the figure, it is difficult to interpret its meaning. If this figure is designed to guide the reader through the data analysis process, how each panel (a to f) relates to one another should be described as well (in the legend).

32. All figures: Please label x and y axes, please include units when applicable (e.g., hormone values). If using a delta score (such as in Figure 3), please indicate that in the axis label.

Reviewer: 2

Comments to the Author

The present manuscript focus on the variation of topological configuration of brain networks across the menstrual cycle, finding an increased betweenness centrality in the right posterior cingulate gyrus during the peri-ovulatory phase. This was related to improved subjective well-being and increased estradiol levels. The purpose of the study is relevant, and especially when the literature on the topic still scarce. MEG and statistical analysis are sound and in general it is well-written. I know longitudinal studies are hard to design and acquire, and I want to compliment the authors on doing so.

However, my main concern is with the experimental design, and I miss some essential information to draw any conclusion related to the cycle phase, which is the main focus of the study. It is indicated that for the ovulatory phase the appointment was scheduled from the 13th to the 15th day of the cycle, and for the luteal, days 21st to 23th, using self-reports. Blood collection was used to measure estradiol, progesterone, LH and FSH levels. It is however not indicated if any cut-off value for hormones or any other index (confirmation of ovulation, confirmation of next period) was used to confirm the cycle phase, or which exclusion criteria was used for participants with a mismatch between the actual and expected hormonal profile (progesterone not highest during the luteal phase or estradiol highest during menses, for example). These kind of longitudinal studies require meticulous experimental design and acquisition and self-reports are not that reliable. Commonly, a single measurement alone is insufficient for cycle phase determination (Becker et al., 2005; Poromaa and Gingnell, 2014). Furthermore, and if I understood it correctly, no statistical analyses were done in order to confirm that women were in the expected cycle phase. Usually, a comparison between the values of the hormone levels in each cycle phase is needed in order to assess a significant endogenous variation (i.e. ANOVA, or lme, ideally taking into account the within subject variation). Given that they already assessed hormonal values, it should be quite straight forward to add an analysis with hormonal values as dependent variables and a summarizing table at least providing mean and SD of each hormonal measurement for each cycle phase. Related to this, I suggest to change the

nomenclature "ovulatory" to "late follicular" or "peri-ovulatory", since it is not assured that they were ovulating at that moment (in any case, if it was meant to capture the pre-ovulatory peak of estradiol, they should refer to it as "pre-ovulatory").

I have other minor suggestions that I hope can help and improve the manuscript. In general, over the introduction and discussion some statements need to be referenced. Specifically in each section:

-Abstract:

Line 28: I would not say that the menstrual cycle is "the only sex hormone-related phenomenon that repeats itself cyclically", as it is known that for example, that testosterone has a diurnal cycle in men, and a monthly cycle is still in question. I suggest to rephrase this sentence.

Line 52: I think the authors mean "later" instead of "letter".

-Introduction:

Line 73: Although here the authors refer to "gender differentiation" in the brain, later they refer to "sexual differentiation". I would say that both refer to sexual differentiation, but in any case, it needs to be consistent.

Line 85: same as for the abstract, line 28.

Line 91: I suggest changing the sentence to "A large number of women suffer from sex hormone-related mood disorders", given that PMDD is not only characterized by depression like symptoms.

Line 107: I suggest to rephrase (or remove) the sentence "However, it has not been possible to identify specific brain locations responsible for higher cognitive function."

Line 113: I would not say that graph theory is typical, but instead, one of the several methods that is used. This last part of the introduction would improve its readability if it were a little more organized. Different analyses (ICA, graph theory) are mixed. For resting state fMRI menstrual cycle analysis and summarized results of current literature I suggest to check also (Hidalgo-Lopez et al., 2020). Given the sparsity of MEG menstrual cycle research, I think Hwang et al., (2008) results need to be described in the introduction.

Line 158: "by invoking", do the authors mean "interpreting"?

-Methods:

Line 173: age and years of education.

Line 182: Do they mean "was over"? "dropped below" doesn't make sense if the cut off was "below" 10 and 21.

Line 183: sample, instead of simple

Line 194: hormone levels

-Statistical analyses and Results:

It is not clear to me exactly what the positive and negative values of the hormonal values differences refer to. Being T1 early follicular, T2, late follicular and T3 luteal, I would expect T1-T2 (as stated in the manuscript) being negative values for estradiol, FSH and LH. In view of the results, I guess the authors meant T2-T1, instead. Please, clarify this.

-Discussion:

I think the discussion is too extensive and could be shortened. Instead of referring to studies already described in the introduction, I think it would benefit from a more detailed explanation of conclusions from the present results. For example, I think the psychological measurements used as index of well-being and the domains that showed significant results are not detailed enough. In any case, is difficult to draw conclusions related to cycle phase when I am not sure all women assigned to each cycle phase were accurately classified. Although results related to correlation between different measurements and estradiol levels should remain as they are now, I'd like to know whether any participant would be excluded when applying a criteria for hormonal values in each phase.

Given that functional asymmetries are considered important to interpret the results I suggest to include some reference to the literature on the neuromodulatory effects of sex hormones on them (see Hodgetts and Hausmann, 2018, for example). Regarding the LOOCV analysis, maybe they can also reference to Franke et al., (2015) study, in which structural changes have already been shown to accurately classify the cycle phase using a machine learning approach, and related to estradiol levels.

Becker, J.B., Arnold, A.P., Berkley, K.J., Blaustein, J.D., Eckel, L.A., Hampson, E., Herman, J.P., Marts, S., Sadee, W., Steiner, M., Taylor, J., Young, E., 2005. Strategies and methods for research on sex differences in brain and behavior. *Endocrinology*. <https://doi.org/10.1210/en.2004-1142>

Franke, K., Hagemann, G., Schleussner, E., Gaser, C., 2015. Changes of individual BrainAGE during the course of the menstrual cycle. *Neuroimage*. <https://doi.org/10.1016/j.neuroimage.2015.04.036>

Hidalgo-Lopez, E., Mueller, K., Harris, T.A., Aichhorn, M., Sacher, J., Pletzer, B., 2020. Human menstrual cycle variation in subcortical functional brain connectivity: a multimodal analysis approach. *Brain Struct. Funct.* 225, 591–605. <https://doi.org/10.1007/s00429-019-02019-z>

Hodgetts, S., Hausmann, M., 2018. The neuromodulatory effects of sex hormones on functional cerebral asymmetries and cognitive control: An update. *Zeitschrift fur Neuropsychol.* 29, 127–139.

<https://doi.org/10.1024/1016-264X/a000224>

Hwang, R.J., Chen, L.F., Yeh, T.C., Tu, P.C., Tu, C.H., Hsieh, J.C., 2008. The resting frontal alpha asymmetry across the menstrual cycle: A magnetoencephalographic study. *Horm. Behav.* 54, 28–33. <https://doi.org/10.1016/j.yhbeh.2007.11.007>

Poromaa, I.S., Gingnell, M., 2014. Menstrual cycle influence on cognitive function and emotion processing from a reproductive perspective. *Front. Neurosci.* 8, 380. <https://doi.org/10.3389/fnins.2014.00380>

Authors' Response

Functional brain network topology across the menstrual cycle is estradiol dependent and correlates with individual well-being

Marianna Liparoti[^], Emahnuel Troisi Lopez[^], Laura Sarno, Rosaria Rucco, Roberta Minino, Matteo

Pesoli, Giuseppe Perruolo, Pietro Formisano, Fabio Lucidi, Giuseppe Sorrentino*, Pierpaolo Sorrentino

Paper ID: jnr-2020-Nov-9204 (REX-PROD-1-84177B9B-0D08-4584-9CE5-4B6FA5472DC5-B0600687-638D-41BD-81D5-C07144CDB342)

Point by point response to the editors' and reviewers' comments.

Review timeline: Submission date: 07/04/2021

Dear

Professor Inger Sundström Poromaa
(Associate Editor, Journal of Neuroscience Research)

and

Dr Junie Warrington
(Editor-in-Chief, Journal of Neuroscience Research)

please find enclosed a revision of our paper entitled "*Functional brain network topology across the*

menstrual cycle is estradiol dependent and correlates with individual well-being" to be considered

for publication in Journal of Neuroscience Research.

We thank the editor and the reviewers for their efforts in evaluating our manuscript.

We did our best to follow the comments made by the reviewers, and hope that this revised submission will be adequate for publication.

In what follows, we firstly summarize the main changes, and then provide detailed answers to Reviewers comments.

Description of major changes

Title. The title has been slightly modified to emphasize the role of estradiol.

Abstract. The abstract has been modified. The hypothesis from which the work starts, and the way

to test it, have been explained in more detail. The results were presented following the criterion adopted in the present modified version (see Methods). Purely speculative hypotheses, such as those

relating to "reproductive psychology", have been eliminated.

Introduction. The introduction has been almost entirely rewritten. In particular, the second paragraph, on the effects of sex hormones throughout the lifespan, has been shortened. The fourth

paragraph, which dealt with psychopathological manifestations in the course of the menstrual cycle

(premenstrual dysphoric disorder and premenstrual syndrome), at the suggestion of Rev 1, was integrated with the literature reporting the occurrence of premenstrual symptoms without clinical relevance. Paragraphs 5 and 6 have been added to address the topic of self-esteem, well-being and

the related assessment tests. In the 5th paragraph, the rationale of examining self-esteem with the Rosenberg test, during the menstrual cycle is clarified. In the sixth paragraph, the Ryff test is introduced, as well as its use in this context. This approach overcomes the paradigm of well-being

as the absence of depressive and/or anxious symptoms, and allows a more thorough description. The paragraphs about sex hormones influencing brain changes, and those on the methodological issues, have been eliminated or condensed. In the last paragraphs, only a few changes have been made.

Methods. Minor changes have been made. In section 2.2 (Experimental protocol) it was described

in details how the phases of the cycle were estimated, namely both as a function of what was reported by each subject, as well as of the hormone levels during the menstrual cycle, mainly the peri-ovulatory and mid luteal phases. The section 2.11 (Statistical analysis) has been modified.

The

correlation analyses were carried out separately in the follicular and luteal phases. In particular, the

difference between the values at T1 and T2 for the follicular phase, and those at T2 and T3 for the

luteal phase, were calculated.

Results. The results of the multilinear model analysis have been moved from paragraph 3.5 to 3.2,

in order to drive the following correlation analyses. Following Rev. 1 suggestion, the correlation analyses were carried out separately taking into account the follicular (\square T1-T2) and luteal (\square T2-T3) phases. These results were described in the corresponding paragraph. These new results highlighted that, during the luteal phase, only the estradiol correlated with both the topological parameter and well-being (environmental mastery domain). Furthermore, following the suggestion

from Rev. 2, the result concerning the comparison of the hormones blood levels among the three time points have been added (see Table 2).

Discussion. Following the input from both Reviewers, the discussion was shortened. The hypothesis

of the topological modification of the posterior cingulate cortex in the peri-ovulatory phase being adaptive has been proposed, shortly, as purely speculative. Finally, a paragraph on the limitations has been added.

RESPONSE TO REVIEWER #1

The authors wish to thank the Reviewer for the criticisms and suggestions, which we have always

found pertinent and constructive.

Major concerns

1.a. Reviewer. Why were the questionnaires Rosenberg Self-Esteem Scale and the Ryff's Test chosen, especially the former? It would be helpful if the Introduction reviewed more previous literature related to specific changes in self-esteem and emotional well-being.

1.a. Authors. Two specific paragraphs have been added to the introduction, explaining the dynamic of self-esteem and well-being (and the related tests) in the context of the menstrual cycle. With regard to low self-esteem as a possible manifestation of subclinical depression, the following text has been added (lines 94-103): *“The association of low self-esteem and anxiety, depression and psychiatric symptoms in general is well known (Paxton et al., 2006). Self-esteem enable subjects to cope more effectively with stressful events (Taylor et al., 2008). Several authors suggest that low self-esteem may be regarded as a subclinical manifestation of depressive mood (Dedovic et al., 2014; Leary et al., 1995; Orth et al., 2008). The association between MC and self-esteem has been proposed. It was found that self-esteem was lower in the premenstrual period in women with premenstrual syndrome (Bloch et al., 1997; Taylor, 1999). Hill and Durante found, in a population of students, that self-esteem was higher during the late luteal phase (Hill and Durante, 2009). However, in a population with similar characteristics, Edmonds et al. failed to find an association between self-esteem and MC (Edmonds et al., 1995).”*

The rationale of choosing the well-being test is explained in the following paragraph (lines 104-116): *“Research on the emotional condition across the MC has been based on the notion of wellbeing, defined as the absence of depressive and/or anxious symptoms. However, evidence suggests that a positive psychological condition entails more than the mere absence of negative mood (Clark and Watson, 1991). Accordingly, the idea that the psychological well-being must be framed in a multidimensional construct has gained acceptance (Kahneman, 1999). In particular, recent studies argue that well-being is best described taking into account elements encompassing both the “hedonic” and the “eudaimonic” dimension, the former being associated with pleasure/displeasure perceptions and changing along with life experiences (Headey, 2006), the latter being associated with homeostatic mechanisms and relatively stable. For an exhaustive review see (Ryan and Deci, 2001). Ryff (Ryff, 1989) claimed that the multidimensionality of the concept of psychological wellbeing can be conceptualized in six core dimensions (i.e. autonomy, environmental mastery, personal growth, positive relation with others, purpose in life and self-acceptance) and developed scales to measure accordingly (Ryff, 2014).”*

1.b. Reviewer. The authors state that the knowledge on menstrual cycle-related changes in emotional experiences (below clinical threshold) are “sparse and anecdotal” -- but there is existing literature that should be discussed.

1.b. Authors. The following sentences have been added (lines 88-93); *“However, a multitude of women complain of cycle-related emotional symptoms. Tschudin et al. reported that 57% of the*

women of childbearing age experience a mild degree of “anger/irritability” or “tearfulness/mood swings” in the premenstrual phase (Tschudin et al., 2010). According to several authors, the prevalence of at least one premenstrual symptom during the woman reproductive life may be much higher, reaching up to 90% (Dennerstein et al., 2012).”

1.c. Reviewer. If space is limited, both of the paragraphs (brain changes across the lifespan and explanation of methods) could be condensed.

1.c. Authors. The paragraphs have been deleted or shortened. Overall, the introduction is now more concise, and we find it easier to read.

2. Reviewer. I believe Spearman's correlation assumes that all observations are independent from one another. This makes it a little more difficult to interpret. Why don't the authors investigate these correlations by menstrual cycle phase?

2. Authors. We changed the statistical analysis. To make interpretation straight forward, the correlation analyses were performed separately across the follicular (T1-T2) and luteal (T2-T3) phases. This way, each observation is independent. The results essentially confirm the trends observed with the previous analysis, although the luteal phase seems to play a prominent role. Specifically, we modified the flow of the analysis, and started the analysis with the linear model. Then, we carried out a correlation analysis between the topological parameters and the hormones blood levels.

3. Reviewer. The Discussion section would benefit from having a clear limitations section. The authors could also include any limitations regarding the study sample or methods used.

3. Authors. The following paragraph has been added (lines 556-567) “*Some limitations of our study deserve attention. First, although the presence of depression and/or anxiety was ruled out using the BDI and the BAI, the lack of premenstrual symptoms was verified only through targeted*

clinical investigation by an experienced gynaecologist, and not with structured questionnaires. Secondly, we estimated the day of ovulation according to backward counting, and we checked hormone levels during the peri-ovulatory phase. However, a single hormone measurement may not

be sufficient to accurately determine the phase of the cycle (Becker et al., 2005; Poromaa and Gingnell, 2014). To be more accurate, one should perform MEG recordings, hormones assays and

psychological assessment daily, and then define the registration corresponding to the pick of LH, but this would not be feasible. Third, it should be considered that the sample was drawn a population with high socio-cultural level, young age, and consisted almost entirely of nulliparous.

Finally, the source reconstruction of two subjects was performed by using a standard template, which might yield less accurate results.”

Minor concerns

4. Reviewer. The second “the” in the title is not necessary.

4. Authors. We modified the title.

5. Reviewer. Abstract: the authors mention in the Abstract (as well as in the main text) that the

menstrual cycle is the only sex hormone-related phenomenon that repeats itself cyclically. This wording should be adjusted, as there are other examples of this.

5. Authors. We agree that this sentence can generate misunderstandings, therefore it has been modified both in the abstract and in the main text (lines 28-29 and 72-75).

6. Reviewer. Abstract: After reading the discussion, I understood what the authors meant by “reproductive psychology” in the abstract. However, I believe this is too complicated a term/argument to be used without explanation. The authors should either explain what they mean by this in regards to their findings, or remove it from the abstract.

6. Authors. We thank the reviewer for this remark that helps the readability of the manuscript. This has now been removed from the abstract.

7. Reviewer. Abstract: It would be informative to mention what aspects of well-being that they analyzed/found significant. The methods should mention the psychological questionnaires used in the study. Otherwise, main parts of the study’s analyses (for example, the self-esteem questionnaire) go unmentioned.

7. Authors. The abstract was modified according to the Reviewer’s comment, clarifying the psychological questionnaires that we used and the results obtained, see lines 36-37 and 41-42 “*psychological questionnaires to quantify anxiety, depression, self-esteem and well-being*” and “*the variation of estradiol correlates positively with the variations of both the topological change and environmental mastery dimension of the well-being test*”.

8. Reviewer. Graphical abstract: Please label all x and y axes. Please state N (perhaps in title, after “Alpha Band”). In the legend, PCG should be rPCG (as in axes). Without reading the manuscript, the reader does not know what T1, T2, or T3 means. It may be clearer to actually label the axis as follicular, ovulatory, and luteal in the graphical abstract.

8. Authors. We modified both the figure and the legend of the graphical abstract.

9. Reviewer. Significance statement: The text may require another proofread.

9. Authors. The manuscript has been carefully revised and the “Significance statement” has been changed.

10. Reviewer. Introduction: The authors hypothesize that brain network topology rearranges across the menstrual cycle as a function of hormone levels, and that these changes are associated with changes in affective condition. If applicable, can the authors be more specific (for example, would they expect a positive/negative correlation between changes in hormones and changes in BC?). I ask because this is a very rich study with a lot of measures (looking at whole brain, all frequency bands, multiple sex hormones, different questionnaires, etc..).

10. Authors. Since this is an exploratory and preliminary study, we did not have ground to make more specific predictions. The role of sex hormones, both on the brain network topology and the

psychological conditions, during the menstrual cycle, has been poorly investigated through MEG.

To the best of our knowledge, only one study, by Hwang et al. (Hwang, R.J., Chen, L.F., Yeh, T.C.,

Tu, P.C., Tu, C.H., Hsieh, J.C., 2008. The resting frontal alpha asymmetry across the menstrual cycle: A magnetoencephalographic study. *Horm. Behav.* 54, 28–33. [PubMed: 25057823]), explores this topic through MEG. However, the regions of interest and the parameters investigated

were different. Therefore, it is quite difficult to hypothesize specific changes of topological parameters a priori.

11. Reviewer. Introduction: Citations are required for lines 70-73 (“Hormonal modulation... differentiation”) and 111-115 (“In the last... by such areas”).

11. Authors. Since these parts were removed, the citations are no longer needed.

12. Reviewer. Introduction: Just checking if “vegetative” is the correct terminology (line 88)?

12. Authors. We changed the term “vegetative” with “neuro-vegetative”, which should be clearer

(please see line 76).

13. Reviewer. Introduction: BOLD is never defined, and fMRI and EEG are defined twice.

13. Authors. This part has been removed in the context of a more thorough editing of the introduction.

14. Reviewer. Introduction: Line 139 is missing something: “...with the change negatively correlated to the (?) and estradiol levels, suggesting that the latter...”

14. Authors. We modified the sentence. Please see lines 137-141 “*Brötzner et al. (Brötzner et al.,*

2014) have associated the alpha frequency oscillations with the MC phases and the hormones levels, and found that the power in the alpha frequency peaked during the luteal phase and the lowest alpha power during the follicular phase, with the change negatively correlated with the estradiol levels, suggesting that the latter modulates the resting state activity in the alpha band.”.

15. Reviewer. Introduction: Line 141 - a null finding is not enough justification for something being “poorly understood.” If there are mixed findings or limited research on the topic, then this should be stated.

15. Authors. The introduction has been almost entirely rewritten and this part has been explained in

further details. The sentence “poorly explored” was removed.

16. Reviewer. Methods: Were there any other exclusion criteria, such as medication use (especially

that affects the CNS), tobacco/alcohol use, or positive drug tests?

16. Authors. We apologise for omitting this information. However, before MEG recordings we routinely collect information regarding the use of drugs or substances such as tobacco, alcohol, coffee, beverages containing caffeine, that could affect the central nervous system. For this study,

we inquired for medicines affecting the menstrual cycle as well as. We modified the inclusion criteria of the study as described below “*We included women who did not make use of hormonal contraceptives (or other hormone regulating medicaments) during the last six months before the recording and who had not been pregnant in the last year. Furthermore, they did not used*

habitually drugs or medicine which could affect the central nervous system and they did not consumed alcohol, tobacco and/or coffee, forty-eight hours prior to the MEG recordings.”

(please

see lines 170-174).

17. Reviewer. Methods: The back-counting method (line 192) needs to be briefly explained, and a citation provided.

17. Authors. To explain the backward counting method, the following sentence was inserted and the references provided: *“We annotated the self-reported last menstrual period, the MC length and*

the date of the predicted onset of the next menses. To achieve greater accuracy in the estimation of

the ovulation, the backward-counting method was applied. This is an indirect counting method that

estimates ovulation by subtracting 14 days from the next predicted period onset. The date of the next period was then confirmed in all included participants (Dixon et al., 1980; Gildersleeve et al.,

2013).” (please see lines 190-194).

18. Reviewer. Methods: What is the “specific questionnaire” for drowsiness?

18. Authors. The Karolinska sleep questionnaire (Åkerstedt, T., Hume, K.E.N., Minors, D., Waterhouse, J.I.M., 1994. The subjective meaning of good sleep, an intraindividual approach using

the Karolinska Sleep Diary. *Percept. Mot. Skills* 79, 287–296. [PubMed:7991323]) was administered. The data was added in the text (please see lines 254-256).

19. Reviewer. Methods: Section 2.9 begins by stating MR images were acquired from 24 participants. As two participants refused to do the MRI scan, shouldn't this be 22?

19. Authors. We apologise for the inaccuracy. The inconsistency was clarified. The text was changed as follows: *“From the total of twenty-four women recruited, twenty-two performed the RM*

while two subjects refused it and a standard template was used for source reconstruction.”

(Please see lines 314-316).

20. Reviewer. Methods: As Section 2.10 describes the psychological evaluations, it would be helpful if the 6 subdomains that were analyzed were mentioned in this section.

20. Authors. In section 2.10 (and in the introduction) we described the six dimensions of wellbeing

of Ryff's test. Furthermore, according to Ruini et al. (*Riv. Psichiatri.* 38, 117, 2003.) in the whole manuscript, we modified the term “subdomains” with “dimensions”.

21. Reviewer. Methods: Apologies that I am not so familiar with MEG recordings, is it standard to

have a break within a 7-minute recording (i.e., at the 3.5 minute mark)?

21. Authors. We thank the Reviewer for allowing us to clarify our methods. In our experimental protocol, the registration length was standard for each participant and for all menstrual cycle phases.

We clarified at lines 248-250 that *“The length of the recording was a trade-off between the need to*

have enough cleaned temporal series, and to avoid drowsiness (Fraschini et al., 2016; Gross et al.,

2013)”. Furthermore, our recording duration is in agreement with Gross et al. (Gross, J., Baillet, S.,

Barnes, G.R., Henson, R.N., Hillebrand, A., Jensen, O., Jerbi, K., Litvak, V., Maess, B., Oostenveld, R., Parkkonen, L., Taylor, J.R., van Wassenhove, V., Wibral, M., Schoffelen, J.M., 2013. Good practice for conducting and reporting MEG research. *Neuroimage* 65, 349–363. [PubMed: 23046981]) in which it is recommended to perform at least two minute of resting state records.

22. Reviewer. Results: For the sake of length, methods descriptions could be removed from the Results section.

22. Authors. We modified the respective sections (3.1, 3.2, 3.3, 3.4, 3.5) of the manuscript according to Reviewer’s suggestion.

23. Reviewer. Results: Negative findings should also be reported in the Results section (for example, the self-esteem questionnaire is not mentioned in the entire Results section, LH and FSH

are not mentioned in Section 3.4).

23. Authors. In all sections, where applicable, we reported negative results.

24. Reviewer. Results: Given that BC rPCG correlates to estradiol, BC rPCG correlates to environmental mastery, and estradiol relates to environmental mastery, couldn’t this be driven by multicollinearity?.

24. Authors. It is a very interesting objection. In order to exclude this possibility, the Variance Inflation Factor analysis was applied to the three parameters. No multicollinearity among the BC of

the rPCG, estradiol blood levels, and environmental mastery scores was showed. We added the appropriate information within the paper as reported below “*Finally, we checked if the relationship*

between the BC of the rPCG, the estradiol levels and the environmental mastery scores could be driven by multicollinearity. The Variance Inflation Factor (VIF) (Belsley et al., 2005; Sneek, 1983)

confirmed that no multicollinearity was present among those three elements (VIF values: BC rPCG

= 1.31, estradiol = 1.95, environmental mastery = 1.91).” (See lines 416-420).

25. Reviewer. Discussion: The “frequently observed mood changes” should be briefly reviewed (or

at least citations included), particularly regarding the measures of interest in this study (e.g., selfacceptance, environmental mastery).

25. Authors. Two citations were suggested. Furthermore, mood disorders have been examined more extensively in the introduction, and are further discussed in this section.

26. Reviewer. Discussion: Line 447 - please specify ‘in the alpha band’ when referring to the “significant reduction of the Lf and the Th”.

26. Authors. The frequency band was specified. (Line 587)

27. Reviewer. Discussion: Line 447-460 - It would be interesting if the authors could discuss the role of sex hormones for this finding. Do the authors have any thoughts on how increased levels of

estradiol, FSH, and LH may help facilitate this (perhaps from a mechanistic standpoint)?

27. Authors. The Reviewer asks how increased levels of hormones may "facilitate" the changes of

the global topological parameters. Unfortunately, this data is phenomenological in nature, and does

not allow any inference on potential biological mechanisms. While some previous knowledge might

suggest some potential pathways (e.g. expression of sex hormone receptor in the central nervous system), in our opinion this is not enough to provide statements on the mechanisms.

Future studies will have to address this crucial point specifically.

28.a. Reviewer. Discussion: Making an evolutionary argument for changes in brain network topology in the ovulatory phase must be done so cautiously. I suggest that the authors make it clearer that this is a highly speculative perspective.

28.a. Authors. We certainly agree with the Reviewer. What is proposed is exclusively a highly speculative hypothesis, albeit, we believe, a fascinating one. It was clearly stated that: "*Albeit within a purely speculative perspective, we notice that*" (Lines 502-504). Later we re-affirmed:

"Some studies are in line with this otherwise speculative hypothesis..." (Lines 509-511).

28.b. Reviewer. The sentence starting on line 499 ("Furthermore") requires citations.

28.b. Authors. The citation (Proverbio et al., 2011) was moved at the end of sentence because it it

refers to both statements.

28.c. Reviewer. I believe the authors are citing Rupp et al., 2009. If this is the case, then it is necessary to mention that that study was conducted in heterosexual women.

28.c. Authors. It was specified that the study was conducted on heterosexual woman. (See line 518)

28.d. Reviewer. If the argument is that this is related to reproduction, I would suggest providing more evidence for this being something specifically related to potential mates/sex/etc. (versus a general increase in affect/social behavior).

28.d. Authors. We believe that, precisely in consideration of the fact that the hypothesis we propose is only speculative, providing further arguments, not supported by objective data, would be

a misleading element for the reader. Furthermore, it would conflict with the attempt to shorten the

text. However, we added the following sentence: "*Further investigation is prompted to understand*

if the observed behavioural changes are specifically related to mating or, rather, a more aspecific

increase toward social engagement." (See lines 522-524).

29. Reviewer. General comment: While sometimes appropriate, authors should be careful using causal, non-flexible language (e.g., "prove", "provoke", "predict").

29. Authors. We have checked the presence of non-flexible language throughout the text, and we

limited using of several words only when strictly appropriate. Specifically, we did an effort to remove any implication of causation where unjustified. However, we kept the use of the word "predict" because it is intended in the statistical sense and refers to a predictive model.

30. Reviewer. Table: A table would be helpful (listing participant information)

30. Authors. A synthetic table containing the requested information (age, cycle length, hormone values per time-point, etc.) was added.

31. Reviewer. Figure 1: Are the images in Figure 1 just representative or based on the data? this figure requires more work.

31 Authors. Figure 1 summarizes the MEG data processing, from acquisition, all the way to topological analysis. According to the Reviewer's suggestion, the figure has been redesigned and the legend perfected, in order to make the explanation clearer. We hope that both of them will be easier to understand now.

32. Reviewer. All figures: Please label x and y axes, please include units when applicable (e.g., hormone values). If using a delta score (such as in Figure 3), please indicate that in the axis label.

32. Authors. All suggestions were accepted, and figures changed accordingly.

RESPONSE TO REVIEWER #2

The authors thank the Reviewer for the attention with which he has examined our work and for the valuable suggestions.

Major concerns

1.a. Reviewer. However, my main concern is with the experimental design related to cycle phase. It is however not indicated if any cut-off value for hormones or any other index (confirmation of ovulation, confirmation of next period) was used to confirm the cycle phase, or which exclusion criteria was used for participants with a mismatch between the actual and expected

hormonal profile (progesterone not highest during the luteal phase or estradiol highest during menses, for example).

1.a. Authors. We tried to explain better the experimental design. In order to define the optimal days

to perform the blood collections and the MEG recordings, we used backward counting, a selfreported

method. Furthermore, we assayed the hormonal blood levels at the three time points, including the estradiol and progesterone during the peri-ovulatory and the mid luteal phase, in order

to confirm that the values were within the normal range. The following sentence was inserted in the

Experimental protocol section "*We annotated the self-reported last menstrual period, the MC length and the date of the predicted onset of the next menses. To achieve greater accuracy in the estimation of the ovulation, the backward-counting method was applied. This is an indirect counting method that estimates ovulation by subtracting 14 days from the next predicted period onset. The date of the next period was then confirmed in all included participants (Dixon et al., 1980; Gildersleeve et al., 2013). Moreover, all the participants included in the study had normal hormonal blood levels (according to the local reference values, reported below) at the three time points, including estradiol and LH in the peri-ovulatory phase and progesterone in the mid luteal phase*". (Lines 190-197).

1.b. Reviewer. These kind of longitudinal studies require meticulous experimental design and acquisition and self-reports are not that reliable. Commonly, a single measurement alone is insufficient for cycle phase determination (Becker et al., 2005; Poromaa and Gingnell, 2014).

1.b. Authors. We are aware that using hormone assay allows us to identify if the subject had an

ovulatory cycle, but we cannot be sure that we performed the MEG recording at the estradiol peak.

The only way would have been to perform daily blood collections for each subject. We highlighted

this point as a limitation in the discussion (See lines 559-562) In any case, it must also be said that

we did not aim to perform the MEG recording (and the psychological evaluation) at the ovulatory

peak, rather to correlate the topological data with the hormonal trend, so much so that we used the

difference of the values between the time points (T1-T2 and T2-T3).

1.c. Reviewer. Furthermore, and if I understood it correctly, no statistical analyses were done in order to confirm that women were in the expected cycle phase. Usually, a comparison between the

values of the hormone levels in each cycle phase is needed in order to assess a significant endogenous variation (i.e. ANOVA, or lme, ideally taking into account the within subject variation). Given that they already assessed hormonal values, it should be quite straight forward to

add an analysis with hormonal values as dependent variables and a summarizing table at least providing mean and SD of each hormonal measurement for each cycle phase.

1.c. Authors. We thank the Reviewer for the very useful suggestion. A statistical analysis (ANOVA) among the three time points, for each hormone, was carried out and the results are presented in Table 2, reported below. Here, we also report the boxplots of the blood levels of the four hormones at the three time points. The latter are not displayed in the text.

Table 2. Sex hormones assay

Sex hormones blood levels (N = 24)

Early follicular Peri-ovulatory Mid-luteal p value†

LH (mIU/ml) $5.4 \pm 2.3^*$ $16.1 \pm 11.8^{**}$ $6.0 \pm 4.0 <0.001$

FSH (mIU/ml) $7.3 \pm 1.4^*$ $7.7 \pm 3.0^{**}$ $3.9 \pm 1.2^{***} <0.001$

Progesterone (ng/ml) $0.3 \pm 0.1^*$ $1.1 \pm 0.8^{**}$ $5.7 \pm 2.6^{***} <0.001$

Estradiol (pg/ml) $33.9 \pm 12.1^*$ $134.3 \pm 70.6^{**}$ $97.4 \pm 39.7^{***} <0.001$

Mean concentrations and standard deviations of luteinizing hormone (LH), follicular stimulant hormone (FSH), progesterone and estradiol estimated in twenty-four women (N = 24) at the three

time points of the MC. *Early follicular vs peri-ovulatory time point: *p value* <0.001; **perioovulatory

vs mid luteal time point: *p value* <0.05; *** Early follicular vs mid luteal time point: *p* <0.001; † ANOVA test.

1.d. Reviewer. Related to this, I suggest to change the nomenclature “ovulatory” to “late follicular”

or “peri-ovulatory”, since it is not assured that they were ovulating at that moment.

1.d. Authors. We believe it is a great suggestion. We changed the nomenclature from “ovulatory”

phase to “peri-ovulatory” phase.

Minor suggestions

-Abstract:

2. Reviewer. Line 28: I would not say that the menstrual cycle is “the only sex hormone-related phenomenon that repeats itself cyclically”. I suggest to rephrase this sentence.

2. Authors. We agree that the sentence might be inaccurate. We rephrased the sentence both in the abstract and in the main text. (See lines 28-29 and 72-75).

3. Reviewer. Line 52: I think the authors mean “later” instead of “letter”.

3. Authors. The mistake has been corrected.

-Introduction:

4. Reviewer. Line 73: Although here the authors refer to “gender differentiation” in the brain, later they refer to “sexual differentiation”. I would say that both refer to sexual differentiation, but in any case, it needs to be consistent.

4. Authors. This part has been removed.

5. Reviewer. Line 85: same as for the abstract, line 28.

5. Authors. We edited this sentence.

6. Reviewer. Line 91: I suggest changing the sentence to “A large number of women suffer from sex hormone-related mood disorders”, given that PMDD is not only characterized by depression like symptoms.

6. Authors. The sentence was rephrased changing “mood disorders” with “psychopathological disorders”

7. Reviewer. Line 107: I suggest to rephrase (or remove) the sentence “However, it has not been possible to identify specific brain locations responsible for higher cognitive function.”

7. Authors. The paragraph was removed and the concepts rephrased in a more concise way.

8. Reviewer. Line 113: I would not say that graph theory is typical, but instead, one of the several methods that is used. This last part of the introduction would improve its readability if it were a little more organized. For resting state fMRI menstrual cycle analysis and summarized results of current literature I suggest to check also (Hidalgo-Lopez et al., 2020). Given the sparsity of MEG menstrual cycle research, I think Hwang et al., (2008) results need to be described in the introduction.

8. Authors. This paragraph was completely rephrased and shorted (please see above

“Description

of major changes” and lines from 117 to 141). The work of Hidalgo-Lopez et al., (2020) was reported (Line 130). Hwang et al., (2008) was already present in the Discussion of the previous version. We would like to keep the citation where it was in the previous version of the manuscript.

9. Reviewer. Line 158: “by invoking”, do the authors mean “interpreting”?

9. Authors. We changed the terminology, and the term “assuming” was used.

-Methods:

10. Reviewer. Line 173: age and years of education.

10. Authors. The mistake was corrected. The subjects’ characteristics were shown in Table 1.

11. Reviewer. Line 182: Do they mean “was over”? “dropped below” doesn’t make sense if the cut off was “below” 10 and 21.

11. Authors. We apologise for the mistake that has now been corrected (see lines 179-181).

12. Reviewer. Line 183: sample, instead of simple

12. Authors. Corrected.

13. Reviewer. Line 194: hormone levels

13. Authors. Modified.

-Statistical analyses and Results:

14. Reviewer. It is not clear to me exactly what the positive and negative values of the hormonal values differences refer to. Being T1 early follicular, T2, late follicular and T3 luteal, I would expect T1-T2 (as stated in the manuscript) being negative values for estradiol, FSH and LH. In view

of the results, I guess the authors meant T2-T1, instead. Please, clarify this.

14. Authors. This observation is not entirely clear to us. Probably the negative or positive values of

the two Δ are not clear because they are presented together. In the present version, at the suggestion

of Reviewer 1, in order to assume the independency of all observation, the correlation analysis was

performed separately on the values of the two Δ (T1-T2 and T2-T3). In any case, we want to reiterate that the essential point for us is that the hormone levels were in the normal range for that specific point of the cycle, thus regardless of the value of the Δ (negative or positive) because our

goal was to correlate the different measures with the trend of estradiol blood levels. However, if we

did not understand the point of the reviewer, we apologize, and remain available to further modify

the analyses.

-Discussion

15.a. Reviewer. I think the discussion is too extensive and could be shortened. For example, I think

the psychological measurements used as index of well-being and the domains that showed significant results are not detailed enough.

15.a. Authors. The discussion has been shortened and hopefully made more readable. The results

about the well-being test and the environmental mastery domain have been provided in further details (compatibly with the need to not lengthen the discussion too much).

15.b. Reviewer. In any case, is difficult to draw conclusions related to cycle phase when I am not

sure all women assigned to each cycle phase were accurately classified.

15.b. Authors. Please, refer to previous responses.

15.c. Reviewer. Although results related to correlation between different measurements and estradiol levels should remain as they are now, I'd like to know whether any participant would be

excluded when applying a criteria for hormonal values in each phase.

15.c. Authors. If the hormone values were not within the normal range at one phase of the cycle, the subject was re-recorded subsequently, together with a new psychological evaluation. The following sentence has been added in the text: "*The subjects (N = 6) that did not have hormonal values in the reference range for each phase, were recorded (and tested) again in the subsequent*

cycles.” (see lines 203-205).

15.d. Reviewer. Given that functional asymmetries are considered important to interpret the results

I suggest to include some reference to the literature on the neuromodulatory effects of sex hormones

on them (see Hodgetts and Hausmann, 2018, for example).

15.d. Authors. In the paragraph dealing with the asymmetry of the topological changes, the works

by Hausmann’s group on the possible mechanisms of sex hormones in determining cerebral asymmetry was reported (see lines 460-462).

15.e. Reviewer. Regarding the LOOCV analysis, maybe they can also reference to Franke et al., (2015) study.

15.e. Authors. It is a very appropriate suggestion. The Franke et al work has now been reported (see lines 497-499).

2nd Editorial Decision

Decision Letter

Dear Dr Liparoti^:

Thank you for submitting your manuscript to the Journal of Neuroscience Research. We have now received the reviewer feedback and have appended those reviews below. I'm glad to say that the reviewers are overall very enthusiastic and supportive of the study. They did raise some concerns and made some suggestions for clarification, but I expect that these points should be relatively straightforward to address. If there are any questions or points that are problematic, please feel free to contact me. I will be glad to discuss.

We ask that you return your manuscript within 30 days. Please explain in your cover letter how you have changed the present version and submit a point-by-point response to the editors’ and reviewers’ comments. If you require longer than 30 days to make the revisions, please contact Dr Junie Warrington (jpwarrington@umc.edu). To submit your revised manuscript: Log in by clicking on the link below <https://wiley.atyponrex.com/submissionBoard/1/afc7c98f-d05f-443c-9f39-657b2798efb3/current>

(If the above link space is blank, it is because you submitted your original manuscript through our old submission site. Therefore, to return your revision, please go to our new submission site here (submission.wiley.com/jnr) and submit your revision as a new manuscript; answer yes to the question “Are you returning a revision for a manuscript originally submitted to our former submission site (ScholarOne Manuscripts)? If you indicate yes, please enter your original manuscript’s Manuscript ID number in the space below” and including your original submission’s Manuscript ID number (jnr-2020-Nov-9204.R1) where indicated. This will help us to link your revision to your original submission.)

The journal has adopted the "Expects Data" data sharing policy, which states that all original articles and reviews must include a Data Availability Statement (DAS). Please see <https://authorservices.wiley.com/author-resources/Journal-Authors/open-access/data-sharing-citation/data-sharing-policy.html#standardtemplates> for examples of an appropriate DAS. Please include the DAS in the manuscript as well.

Thank you again for your submission to the Journal of Neuroscience Research; we look forward to reading your revised manuscript.

Best Wishes,

Professor Inger Sundström Poromaa
Associate Editor, Journal of Neuroscience Research

Dr Junie Warrington
Editor-in-Chief, Journal of Neuroscience Research

Associate Editor: Sundström Poromaa, Inger
Comments to the Author:
(There are no comments.)

Reviewer: 1

Comments to the Author

Regarding potential statistical analysis concerns - please see point 8 and point 9.

The authors have done an excellent job addressing the previous comments raised in the review process, especially in the Introduction section and analyses. My remaining concerns, based on the new edits, are mostly minor points.

Abstract:

1. In their previous response letter, the authors said that they cannot be more specific about their hypotheses as this is an exploratory and preliminary study. This is absolutely fair, but then the nature of the study could be briefly mentioned in the abstract (perhaps just specifying "In this exploratory study" (line 29)).

Graphical abstract:

2. "Rep measures" should be defined or removed. Use of either cycle phase names or T2/T3 should be consistent (panel A uses phase names, panel C uses T2/T3). While I prefer cycle phase names, this is at the discretion of the authors.

Introduction:

3. "Menopause" may be a more widely recognized term compared to "Climacteric" (line 70).

4. "Unlike puberty or menopause, which are unique and non-repeatable processes, the menstrual cycle..." (line 72) suggests that puberty occurs at a single time-point and is independent from the menstrual cycle, which is not true.

5. Instead of saying that "women complain of cycle-related emotional symptoms" (line 88), I suggest more neutral language, such as "women report".

6. In several instances, the authors report previous findings according to "several authors". I believe it is more conventional to refer to reviews, studies, etc. - not authors.

7. In the final paragraph, it would be helpful if the authors state which sex hormones they are investigating.

Methods:

8. When assessing hormone changes across cycle phases, the authors initially perform an ANOVA "followed by post-hoc analysis between groups, using Student's t-test" (line 331-332). I believe the authors mean cycle phase when referring to groups? The authors should be using paired sample t-tests (not comparing the means of two independent groups) when comparing the same participants at two different time-points.

9. The authors discuss their findings as changes across cycle phases (e.g., a change between the peri-ovulatory and mid-luteal time-points). Just to confirm, when the authors express the delta as " $\Delta T1-T2$, defined as the luteal phase", are they performing the change score as T2 minus T1? If this is how the analyses were performed, I would instead write $\Delta T2-T1$. If this is not how the analyses were performed, I would calculate the delta as the later time-point minus the earlier time-point, as this would make it easier to interpret and more reflective of changes across a cycle phase. I believe Reviewer 2 already raised this concern.

10. As the authors list the expected profile for estradiol and progesterone for the early follicular and mid-luteal phases, they should do the same for the peri-ovulatory phase (lines 188-189).

11. For the ultrasound examination, was the gathered information only used for detecting healthy/normal uteri and ovaries? If this information was also used for estimating ovulation timing, this should be mentioned in the relevant section, as this is a strength (especially given the variance seen in the LH levels during the peri-ovulatory time-point). Otherwise, please ignore comment.

Results & Table 2

12. It is appreciated that the authors now report the sex hormone levels. But the results section states that, per sex hormone, the levels significantly differ at each time-point from the other two time-points (line 369). As I understand from Table 2, this is not the case for LH in the comparison of early follicular to mid-luteal. While this is not unexpected, the text should be adjusted.

13. The asterisk system in Table 2 is slightly confusing (e.g., three asterisks versus one represents the same p-value but for a different phase comparison). It would be easier to interpret if the p-values for the three phase comparisons were listed in separate columns that actually state which phases are being compared (more informative than the ANOVA p-value, which could just be listed in the text, if space is an issue).

Discussion:

14. The authors state that there is a "strong tendency" (line 434) towards significance for the correlation of BC of the rPCG and estradiol, but $pFDR = 0.453$.

15. In the paragraph starting on line 526, the authors discuss their luteal phase findings in the context of previous work that has shown worse mood symptoms and maladaptive brain responses during the premenstrual phase. For example, in PMDD, symptoms occur shortly before menstruation in the late luteal phase. The authors should briefly acknowledge, however, that their study examined the *mid*-luteal phase.

General:

16. There are still (minor) grammatical issues that the authors could fix. For example, "during the woman

reproductive life" (line 29), "entails more than the mere absence" (line 106), "they did not use... they did not consume" (lines 172-173), etc..

Reviewer: 2

Comments to the Author

I thank the authors for providing a thorough and detailed revision. All my comments have been addressed, so I endorse the manuscript for publication in its current form.

Authors' Response

Functional brain network topology across the menstrual cycle is estradiol dependent and correlates with individual well-being

Marianna Liparoti[^], Emahnuel Troisi Lopez[^], Laura Sarno, Rosaria Rucco, Roberta Minino, Matteo Pesoli, Giuseppe Perruolo, Pietro Formisano, Fabio Lucidi, Giuseppe Sorrentino*, Pierpaolo Sorrentino
Paper ID: jnr-2020-Nov-9204.R1

Review timeline: Submission date: 10 November 2020

Editorial Decision: 17 February 2021

Revision Received: 7 April 2021

Editorial Decision: 4 May 2021

Revision Received: 11 May 2021

Accepted:

Dear

Professor Inger Sundström Poromaa
(Associate Editor, Journal of Neuroscience Research)

and

Dr Junie Warrington
(Editor-in-Chief, Journal of Neuroscience Research)

please find enclosed a revision of our paper entitled "*Functional brain network topology across the menstrual cycle is estradiol dependent and correlates with individual well-being*" to be considered for publication in Journal of Neuroscience Research.

We thank the Editor and the Reviewer for his/her suggestions. We edited the manuscript hoping this revised submission will be adequate for publication in your journal. Below the point-by-point response to Reviewer.

RESPONSE TO REVIEWER #1

Comments to the Author

Regarding potential statistical analysis concerns - please see point 8 and point 9.

The authors have done an excellent job addressing the previous comments raised in the review process, especially in the Introduction section and analyses. My remaining concerns, based on the new edits, are mostly minor points.

Major concerns

Abstract:

1. Reviewer. In their previous response letter, the authors said that they cannot be more specific about their hypotheses as this is an exploratory and preliminary study. This is absolutely fair, but then the nature of the study could be briefly mentioned in the abstract (perhaps just specifying "In this exploratory study" (line 29)).

1. Authors. The abstract has been modified (see line 29).

Graphical abstract:

2. Reviewer. "Rep measures" should be defined or removed. Use of either cycle phase names or T2/T3

should be consistent (panel A uses phase names, panel C uses T2/T3). While I prefer cycle phase names, this is at the discretion of the authors.

2. Authors. We modified the figure and the caption of the graphical abstract as suggested by the Reviewer.

Introduction:

3. Reviewer. “Menopause” may be a more widely recognized term compared to “Climacteric” (line 70).

3. Authors. We modified this term (See line 72).

4. Reviewer. “Unlike puberty or menopause, which are unique and non-repeatable processes, the menstrual cycle...” (line 72) suggests that puberty occurs at a single time-point and is independent from the menstrual cycle, which is not true.

4. Authors. We agree with the Reviewer and we modified this sentence as reported below “Unlike puberty or menopause, which are processes that occur in adolescence and adulthood respectively, the menstrual cycle (MC) is a hormone-related phenomenon that accompanies the woman from puberty to menopause and repeats itself cyclically with periodical and coordinated variations of multiple hormones, such as estradiol, progesterone, Follicular Stimulant Hormone (FSH) and Luteinizing Hormone (LH).” (Please see lines 74-78)

5. Reviewer. Instead of saying that “women complain of cycle-related emotional symptoms” (line 88), I suggest more neutral language, such as “women report”.

5. Authors. We replaced the term “complain of” with “report” as suggested by the Reviewer (see line 91).

6. Reviewer. In several instances, the authors report previous findings according to “several authors”. I believe it is more conventional to refer to reviews, studies, etc. – not authors.

6. Authors. We made the requested change (see lines 94 and 99).

7. Reviewer. In the final paragraph, it would be helpful if the authors state which sex hormones they are investigating.

7. Authors. We added this information in the final part of the introduction (please see lines 164-165).

Methods:

8. Reviewer. When assessing hormone changes across cycle phases, the authors initially perform an ANOVA “followed by post-hoc analysis between groups, using Student’s t-test” (line 331-332). I believe the authors mean cycle phase when referring to groups? The authors should be using paired sample ttests

(not comparing the means of two independent groups) when comparing the same participants at two different time-points.

8. Authors. We modified the 2.11 section of main text specifying that we applied the Anova test to calculate the variance of each hormone (LH, FSH, progesterone and estradiol) estimated in twenty-four women at three points of the menstrual cycle (during early follicular (T1), peri-ovulatory (T2) and midluteal

(T3) phases). Subsequently the post-hoc analysis between cycle time points (T1 vs T2, T2 vs T3 and T1 vs T3) was performed using the paired sample t-test. We modified Table 2 in order to clarify the statistical analysis performed.

9. Reviewer. The authors discuss their findings as changes across cycle phases (e.g., a change between the peri-ovulatory and mid-luteal time-points). Just to confirm, when the authors express the delta as “ $\Delta T1-T2$, defined as the luteal phase”, are they performing the change score as T2 minus T1? If this is how the analyses were performed, I would instead write $\Delta T2-T1$. If this is not how the analyses were performed, I would calculate the delta as the later time-point minus the earlier time-point, as this would make it easier to interpret and more reflective of changes across a cycle phase. I believe Reviewer 2 already raised this concern.

9. Authors. The Reviewer reports our statement: “ ΔT_1-T_2 , defined as the luteal phase”. We checked the test, but we could not find this statement. However, his/her suggestion to calculate the delta as the later time-point minus the earlier time point is quite acceptable as more reflective of the physiological curves of the sex hormones during the menstrual cycle. The test and the figures were changed accordingly.

10. Reviewer. As the authors list the expected profile for estradiol and progesterone for the early follicular and mid-luteal phases, they should do the same for the peri-ovulatory phase (lines 188-189).

10. Authors. We added the requested information (please see line 193).

11. Reviewer. For the ultrasound examination, was the gathered information only used for detecting healthy/normal uteri and ovaries? If this information was also used for estimating ovulation timing, this should be mentioned in the relevant section, as this is a strength (especially given the variance seen in the LH levels during the peri-ovulatory time-point). Otherwise, please ignore comment.

11. Authors. In order to verify the presence of abnormal findings, the ultrasound examination has to be performed in the early follicular phase. According to Ethic Committee, to perform a further ultrasound during the peri-ovulatory phase was too much unpleasant for the participants to the study.

Results & Table 2

12. Reviewer. It is appreciated that the authors now report the sex hormone levels. But the results section states that, per sex hormone, the levels significantly differ at each time-point from the other two time-points (line 369). As I understand from Table 2, this is not the case for LH in the comparison of early follicular to mid-luteal. While this is not unexpected, the text should be adjusted.

12. Authors. Sorry for the mistake. In agreement with the Reviewer’s comment we modified the results paragraph (please see lines 375-379).

13. Reviewer. The asterisk system in Table 2 is slightly confusing (e.g., three asterisks versus one represents the same p-value but for a different phase comparison). It would be easier to interpret if the p-values for the three phase comparisons were listed in separate columns that actually state which phases are being compared (more informative than the ANOVA p-value, which could just be listed in the text, if space is an issue).

13. Authors. We apologize for not being clear in the description of the results. Table 2 has been modified following the Reviewer’s suggestions.

Discussion:

14. Reviewer. The authors state that there is a “strong tendency” (line 434) towards significance for the correlation of BC of the rPCG and estradiol, but $pFDR = 0.453$.

14. Authors. We modified the sentence as reported below “*during the follicular phase a statistical significance was observed before FDR correction*” (please lines 444-445).

15. Reviewer. In the paragraph starting on line 526, the authors discuss their luteal phase findings in the context of previous work that has shown worse mood symptoms and maladaptive brain responses during the premenstrual phase. For example, in PMDD, symptoms occur shortly before menstruation in the late luteal phase. The authors should briefly acknowledge, however, that their study examined the *mid*-luteal phase.

15. Authors. We added the information suggested by the Reviewer as reported below “...(defined as the difference between mid-luteal and peri-ovulatory phases of MC)...” (please see line 561).

General:

16. Reviewer. There are still (minor) grammatical issues that the authors could fix. For example, “during the woman reproductive life” (line 29), “entails more that the mere absence” (line 106), “they did not used... they did not consumed” (lines 172-173), etc..

16. Authors. We revised the manuscript by modifying all grammatical errors.

3rd Editorial Decision

Decision Letter

Dear Dr Liparoti^:

Thank you for submitting your manuscript "Functional brain network topology across the menstrual cycle is estradiol dependent and correlates with individual well-being" by Liparoti^, Marianna; Troisi Lopez, Emahnel; Sarno, Laura; Rucco, Rosaria; Minino, Roberta; Pesoli, Matteo; Perruolo, Giuseppe; Formisano, Pietro; Lucidi, Fabio; Sorrentino, Giuseppe; Sorrentino, Pierpaolo.

You will be pleased to know that your manuscript has been accepted for publication. Thank you for submitting this excellent work to our journal.

In the coming weeks, the Production Department will contact you regarding a copyright transfer agreement and they will then send an electronic proof file of your article to you for your review and approval.

Please note that your article cannot be published until the publisher has received the appropriate signed license agreement. Within the next few days, the corresponding author will receive an email from Wiley's Author Services asking them to log in. There, they will be presented with the appropriate license for completion. Additional information can be found at <https://authorservices.wiley.com/author-resources/Journal-Authors/licensing-open-access/index.html>

Would you be interested in publishing your proven experimental method as a detailed step-by-step protocol? Current Protocols in Neuroscience welcomes proposals from prospective authors to disseminate their experimental methodology in the rapidly evolving field of neuroscience. Please submit your proposal here: <https://currentprotocols.onlinelibrary.wiley.com/hub/submitproposal>

Congratulations on your results, and thank you for choosing the Journal of Neuroscience Research for publishing your work. I hope you will consider us for the publication of your future manuscripts.

Sincerely,

Professor Inger Sundström Poromaa
Associate Editor, Journal of Neuroscience Research

Dr Junie Warrington
Editor-in-Chief, Journal of Neuroscience Research

Associate Editor: Sundström Poromaa, Inger
Comments to the Author:
(There are no comments.)

Authors' Response

4th editorial decision

Decision Letter

Author response